# Transformers in the Dark:
# Navigating Unknown Search Spaces via Bandit Feedback

**Jungtaek Kim**                                                          *jungtaek.kim@wisc.edu*
*University of Wisconsin–Madison*

**Thomas Zeng**
*University of Wisconsin–Madison*

**Ziqian Lin**
*University of Wisconsin–Madison*

**Minjae Lee**
*FuriosaAI*

**Chungpa Lee**
*Yonsei University*

**Jy-yong Sohn**
*Yonsei University*

**Hyung Il Koo**
*FuriosaAI*

**Kangwook Lee**                                                          *kangwooklee@krafton.com*
*University of Wisconsin–Madison, KRAFTON AI, Ludo Robotics*

**Reviewed on OpenReview:** *https://openreview.net/forum?id=Jij7zCjVfc*

## Abstract

Effective problem solving with Large Language Models (LLMs) can be enhanced when they are paired with external search algorithms. By viewing the space of diverse ideas and their follow-up possibilities as a tree structure, the search algorithm can navigate such a search space and guide the LLM toward better solutions more efficiently. While the search algorithm enables an effective balance between exploitation and exploration of a tree-structured space, the need for an external component can complicate the overall problem-solving process. We therefore pose the following question: *Can LLMs or their underlying Transformer architectures approximate a search algorithm?* To answer this question, we first introduce a simplified framework in which tree extensions and feedback signals are externally specified, allowing for controlled evaluation of search capabilities. We call this setting *unknown tree search with bandit feedback*. Within this setting, we show that Transformers are theoretically expressive enough to implement distinct search strategies and can be trained from scratch to approximate those strategies. Our Transformer models exhibit the possibility of generalizing to unseen conditions such as longer horizons or deeper trees. Furthermore, we demonstrate that continued task-focused training unlocks the complete capabilities of a pretrained LLM, by fine-tuning the LLM on search trajectories.[1]

---

[1]Our implementation and training configurations are publicly available at `https://github.com/UW-Madison-Lee-Lab/Transformers-in-the-Dark`.

# 1 Introduction

Effective problem solving often follows an iterative process of (i) generating diverse idea candidates, (ii) selecting the most promising one to pursue, and (iii) evaluating its potential. Large Language Models (LLMs) (Brown et al., 2020; Llama Team, AI @ Meta, 2024; Abdin et al., 2024), including reasoning-focused variants (DeepSeek-AI, 2025; Gemini Team, Google, 2025; OpenAI, 2025b), suggest that these systems may implicitly implement such a process, as visualized in Figure 1a. Moreover, as discussed in previous literature (Yao et al., 2023a; Hao et al., 2023; Zhou et al., 2024), this problem-solving process can be enhanced when LLMs are paired with external search algorithms. By treating the space of diverse ideas and their follow-up possibilities as a tree-structured space, the search algorithm navigates this search space, allowing it to efficiently guide the LLM toward better solutions. While the search algorithm lets the model exploit and explore effectively, relying on an external component can complicate the overall problem-solving process.

We therefore raise the following research question:

*Can LLMs or their underlying Transformer architectures approximate a search algorithm?*

To address this question, we introduce *unknown tree search with bandit feedback*, a simplified framework where search space expansions are provided externally and rewards are given as bandit feedback, as presented in the table shown in Figure 1. By removing self-generated structures, this setup isolates the model's ability to balance exploitation and exploration during selection, enabling controlled evaluation of LLM behavior under uncertainty. Thus, this provides a rigorous foundation for assessing whether LLMs genuinely implement structured search and how their strategies operate in such settings.

Within this setting, we show that, theoretically, Transformers are expressive enough to represent a wide range of search strategies, and empirically, they can be trained from scratch to imitate such strategies and to exhibit the possibility of generalizing beyond the training distribution, for example to longer horizons or deeper trees. However, existing LLMs are still limited in our simple setting, as shown in Figure 1c. We compare them to the uniform and greedy leaf sampling strategies and the variant of Monte Carlo Tree Search (MCTS) (Sutton & Barto, 2018) in terms of max-reward hit rate; see Sections 3.2 and 5 for the details of the search algorithms and the metric, respectively. The LLMs tested in this setting underperform compared to the established search algorithms. In particular, Qwen3-8B is on par with the uniform leaf sampling, which is a naïve search algorithm, while Qwen3-8B Thinking performs even worse.

The observed performance gap indicates that current LLMs still have limited search capabilities, making them less effective as problem-solving agents when search is the core challenge. To fully realize their potential, targeted training is required. Our experiment with fine-tuned Qwen3-8B (Qwen Team, 2025) shows that training specifically designed for search under uncertainty significantly enhances LLM effectiveness compared to an LLM-only method.

We summarize our contributions as follows:

- We introduce unknown tree search with bandit feedback, a simplified and controlled setting that captures the essence of problem solving by externalizing both expansions and feedback;

- We provide a theoretical analysis showing that Transformers are expressive enough to represent distinct strategies and conduct empirical studies demonstrating that Transformers trained with these strategies can perform the approximation of behavior cloning and generalization to unseen conditions;

- We demonstrate that additional training explicitly designed for search under uncertainty improves LLM performance, mitigating the gap relative to specialized algorithms.

# 2 Related Work

In this section, we briefly review the background and related work relevant to this paper.

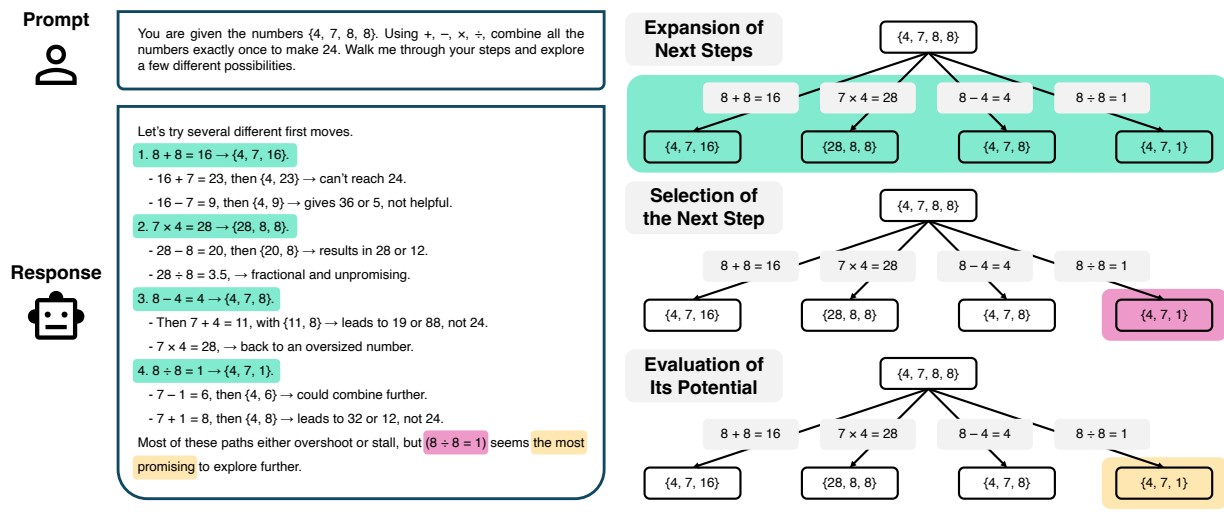

(a) Effective problem solving with an LLM where a prompt on a game of 24 is given

| | LLM-Guided Tree Search | Tree Search with External Search | Unknown Tree Search with Bandit Feedback |
|---|---|---|---|
| Expansion | LLM | LLM | Externally given |
| Selection | LLM | External search | LLM or Transformer |
| Evaluation | LLM | LLM | Externally given |

(b) Comparison of our problem formulation to existing problem formulations

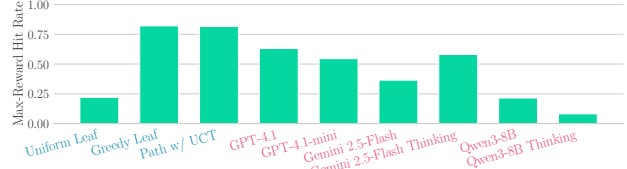

(c) Results of reference search algorithms (blue text) and existing LLMs (red text)

Figure 1: Our perspective on effective problem solving as an iterative process of three phases. Given a prompt that describes a simple problem, i.e., a game of 24, an LLM generates several possible steps, selects the next step, and finally evaluates its potential. Repeating this cycle constructs a tree-structured search space, where each branch represents a potential path of problem solving. This example is generated by GPT-5, and paraphrased to clearly illustrate our definition of effective problem solving. Under this perspective, existing problem formulations and our problem formulation (highlighted in gray) are summarized in Figure 1b. Specifically, our formulation assumes that next step selection is carried out by an LLM or a Transformer while state expansion and state evaluation are externally given. Figure 1c shows the results on multi-reward tree search with binary trees of depth 6 and 8 different goal states; refer to Sections 5 and E.1 for the details of the metric and the experiment, respectively. Existing LLMs are inferior to some of the established algorithms, and Qwen3-8B Thinking is even worse than uniform leaf sampling, which is a naïve strategy.

**Blind and Uninformed Search** Blind or uninformed search refers to algorithms that explore a search space without heuristic guidance or prior knowledge of its structure (Russell & Norvig, 2010). Classical methods such as Breadth-First Search (BFS), Depth-First Search (DFS), and uniform-cost search systematically enumerate nodes, guaranteeing completeness and, in some cases, optimality, but scale poorly in large or sparse environments due to their inability to incorporate feedback during search (Russell & Norvig, 2010). Modern extensions introduce adaptivity via stochastic sampling and online feedback. MCTS (Sutton & Barto, 2018), particularly upper confidence bounds applied to trees (UCT) (Kocsis & Szepesvári, 2006), balances exploration and exploitation by applying bandit principles to tree expansion and has proven highly effective in diverse domains such as Go (Silver et al., 2016) and Atari games (Mnih et al., 2013). Bandit-based tree search has been further studied in the context of regret minimization and sample complexity (Bubeck et al., 2011).

**Learning-Based Planning** It denotes training models to acquire planning abilities from data, allowing them to generate action sequences or policies for solving tasks under uncertainty without hand-crafted search. MuZero learns both dynamics model and search policy directly from interaction, achieving strong

performance without access to game rules (Schrittwieser et al., 2020). Valmeekam et al. (2023a) propose benchmarks for evaluating LLMs on planning and reasoning, and given existing pretrained LLMs such as GPT-3.5 and GPT-4 (OpenAI, 2023), Valmeekam et al. (2023b) analyze their planning abilities. Chen et al. (2021) predict actions given the history of states, actions, and rewards, using the Transformer architecture. In addition, historical episodes are collected from a source RL algorithm and then they are used to autoregressively train a Transformer model (Laskin et al., 2023), which is called algorithmic distillation. Recent studies (Lehnert et al., 2024; Nolte et al., 2024; Su et al., 2025) propose intriguing methods to train the Transformer-based model on the traces of A* search. In particular, the method by Nolte et al. (2024) is capable of controlling Transformers' outputs using either fast or slow mode. This line of research provides the entire environment to the models, unlike our problem formulation. Beyond a framework with reasoning and acting (Yao et al., 2023b), Zhou et al. (2024) make use of the MCTS algorithm in decision-making with LLMs. Kambhampati et al. (2024) introduce the LLM-Modulo framework, in which LLMs generate ideas and serve as external critics.

**Reasoning Abilities of LLMs**  Popular LLMs (DeepSeek-AI, 2025; Gemini Team, Google, 2025; OpenAI, 2025b) enhance its performance using a reasoning mechanism. Lightman et al. (2024) compare outcome-supervised and process-supervised reward models for guiding a pretrained LLM's chain-of-thought reasoning, showing that step-level supervision yields more reliable reasoning. Dziri et al. (2023) find that Transformers encounter limitations in compositional multi-step reasoning, leading to performance degradation as task complexity increases. Sun et al. (2024) show that the reward models trained on easier problems can generalize to supervise harder ones, enabling scalable alignment without direct human feedback. Snell et al. (2025) investigate that allocating inference-time compute effectively can improve LLM performance, comparing larger models under equal budgets. Han et al. (2025) introduce a framework that dynamically considers chain-of-thought token budgets relative to the complexity of each problem, reducing the number of tokens with minimal loss in accuracy. Khalifa et al. (2025) propose ThinkPRM, a generative chain-of-thought process reward model that, with limited step-level supervision, surpasses both LLM-as-a-judge and discriminative PRMs on several benchmarks.

## 3  Problem Formulation and Model Interfaces

Each problem instance is defined by a finite, rooted search tree $\mathcal{T} = (\mathcal{S}, N)$ with maximum depth $D$, where $\mathcal{S}$ is a finite set of states and $N : \mathcal{S} \to 2^{\mathcal{S}}$ is a successor function mapping each state to its children. We assume a bounded reward function $r : \mathcal{S} \to [0, 1]$ and define the set of *goal states* as $\mathcal{S}_{\text{goal}} = \{s \in \mathcal{S} : r(s) > 0\}$, requiring each goal state to be a leaf node in $\mathcal{T}$. Goal states are assumed to be sparse, i.e., $|\mathcal{S}_{\text{goal}}| \ll |\mathcal{S}|$, with most states yielding zero reward, reflecting realistic scenarios where solutions are rare. Importantly, the tree $\mathcal{T}$ constitutes the underlying search space, which remains hidden from the search agent.

We define the *value* of a state $s$ under a uniformly random rollout policy as follows:

$$V(s) = \mathbb{E}_{\pi_{\text{unif}}}[r(s_{\text{leaf}}) \mid s_{\text{start}} = s], \tag{1}$$

where $\pi_{\text{unif}}$ denotes a uniform random traversal policy through the tree, and $s_{\text{leaf}}$ represents the leaf state reached after traversing from the initial state $s_{\text{start}} = s$. To simulate real-world scenarios where exact feedback is unavailable, we approximate $V(s)$ by a random variable $\widehat{V}(s)$. Each realization of $\widehat{V}(s)$ is obtained via a Monte Carlo or heuristic estimate of $V(s)$. For example, the Monte Carlo estimate is generated by performing $k$ independent rollouts using $\pi_{\text{unif}}$. On the other hand, in our experiments with pretrained or fine-tuned LLMs on "real-world" search trees, we adopt a more practical formulation of $\widehat{V}(s)$, which requires internal estimation. Specifically, we can prompt the LLM to directly estimate state values, e.g., by producing a score between 0 and 1. These realizations serve as noisy bandit feedback signals received by the search agent.

Given the above setup, the agent-environment interaction occurs over $T$ steps, where $T \ll |\mathcal{S}|$. Starting from the root state $s_0$, at each step $t$, the environment reveals the children $N(s_t)$ and a sampled rollout value $v_t \sim \widehat{V}(s_t)$. We define the frontier of unvisited child states as follows:

$$F_t = (\bigcup_{i=0}^{t} N(s_i)) \setminus \{s_0, s_1, \ldots, s_t\}, \tag{2}$$

the search agent selects the state for the next step, $s_{t+1} \in F_t$, according to its policy:

$$\pi(\cdot \mid s_0, v_0, N(s_0), s_1, v_1, N(s_1), \ldots, s_t, v_t, N(s_t)). \tag{3}$$

After $T$ steps, we obtain a completed search trajectory $\tau$:

$$\tau = [(s_0, v_0, N(s_0)), (s_1, v_1, N(s_1)), \ldots, (s_T, v_T, N(s_T))]. \tag{4}$$

We evaluate $\tau$ by the reward achieved at its best (highest-reward) state, $r(s_{\text{best}}) = \max_{s_i \in \tau} r(s_i)$. For later convenience, we collect the individual state rewards along the trajectory into the set:

$$\mathcal{R} = \{r(s_i) \mid (s_i, v_i, N(s_i)) \in \tau\}. \tag{5}$$

With this notation, the previous expression can be written simply as $r(s_{\text{best}}) = \max \mathcal{R}$. In Section 5, we provide additional diverse metrics, which are computed as functions of $\mathcal{R}$, to provide a more nuanced understanding of each search strategy.

### 3.1 Specific Instances with Tree-Structured Search Spaces

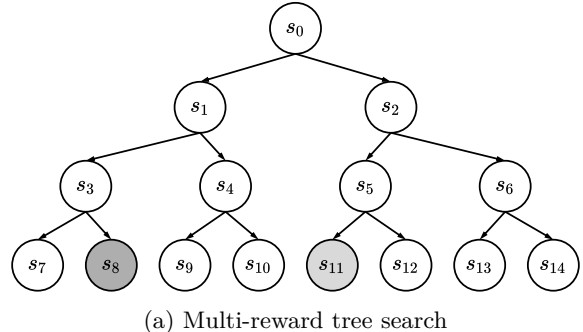

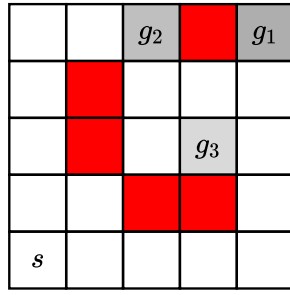

(a) Multi-reward tree search

(b) Multi-reward navigation

Figure 2: Two environments investigated in this work, where darker cells represent higher reward values and red cells denote cells that are impassable.

Given the general setup, we detail two concrete problems with tree-structured search spaces; their illustrations are shown in Figure 2. The settings of these synthetic problems can be easily controlled to adjust their difficulty, which enables us to conduct the analysis of our Transformer models.

**Multi-Reward Tree Search** This problem is a straightforward implementation of our setup, by directly generating a randomized search tree. It is defined with three parameters: (i) a branching factor at each intermediate state $B$; (ii) a tree depth $D$; (iii) the number of goal states $K$. The number of accessible states is $(B^{D+1} - B)/(B - 1)$, where a root state $s_0$ is excluded since it is initially given. All goal states are located in the leaf nodes of the tree. In Figure 2a, $s_0$, $s_1$, ..., $s_{14}$ are all states in this binary tree, and $s_8$ and $s_{11}$ are goal states where $r(s_8) > r(s_{11})$.

**Multi-Reward Navigation** We define a multi-reward maze traversal task within our problem setup. The objective of this task is to find paths from a designated start vertex $v_{\text{start}}$ to one of several goal vertices. Specifically, consider an undirected graph $\mathcal{G} = (V, E)$ representing a two-dimensional grid of size $w \times h$, where $|V| = wh$ and the edges $E$ connect neighboring vertices (i.e., up, down, left, right). Some of $wh$ vertices, which are called *walls*, cannot be traversed. Here, we define a wall density $n_{\text{wall}}/wh$, where $n_{\text{wall}}$ is the number of wall vertices. The graph contains a designated start vertex $v_{\text{start}} \in V$ and a set of goal vertices $G = \{g_1, \ldots, g_k\} \subseteq V$, each associated with a strictly decreasing positive reward:

$$R(g_1) > R(g_2) > \cdots > R(g_k) > 0, \tag{6}$$

and $R(v) = 0$ for all non-goal vertices $v \notin G$. The branching factor $B$ is naturally bounded by 4 due to the grid structure.

To formulate this problem as a search problem within our framework, we define the search space $\mathcal{S}$ as the set of all valid paths $\rho = (v_0, \ldots, v_\ell)$ in $\mathcal{G}$, satisfying the following constraints: (i) the initial vertex is fixed as $v_0 = v_{\text{start}}$, (ii) each path has length at most $T$, and (iii) each path visits at most one goal vertex, which if visited, must be the terminal vertex of the path. The successor function is defined by $N(\rho) = \{\rho \circ v \in \mathcal{S} : v \in V\}$, where $\rho \circ v$ denotes appending vertex $v$ to the path $\rho$. Finally, the reward function is defined as $r(\rho) = R(v_\ell)$, assigning the reward of the terminal vertex $v_\ell$ to the entire path. This construction naturally yields a tree representation suitable for our general search framework; see Figures 2b and 9 for the detailed visualization of this construction.

More specifically, in Figure 2b, an agent is started from a start vertex $s$ and then it can select one of neighbor vertices (i.e., up, down, left, and right vertices of the vertex of interest) sequentially. Red cells indicate impassible vertices. Similar to the binary tree example, $r(g_1) > r(g_2) > r(g_3)$. It is noteworthy that we express states as paths from $s$ to a particular vertex which is shown in Figure 2b, so that it can straightforwardly create a tree-structured search space.

### 3.2 Reference Search Algorithms

Here, we introduce several reference search algorithms to solve the problem discussed above.

**Uniform Leaf Sampling**    This method chooses one of $F_{t-1}$ uniformly at random at step $t$: $s_t \sim \mathcal{U}(F_{t-1})$, where $\mathcal{U}$ is a uniform distribution; refer to Algorithm 2 for details.

**Greedy Leaf Sampling**    This algorithm selects one of $F_{t-1}$ whose parent state has the highest reward value: $s_t = \arg\max_{s \in F_{t-1}} \hat{v}(s)$, where $\hat{v}(s)$ is the estimated value of the parent state of $s$; refer to Algorithm 3 for details.

**Uniform Path Sampling**    This strategy uniformly samples the next node at each depth level starting from a root state $s_0$ until it meets an instance in $F_{t-1}$; see Algorithm 4 for details.

**Policy-Guided Path Sampling**    These methods traverse an underlying tree using one of tree traversal policies: pure exploration, greedy, and UCT policies; see Algorithm 5 for details. It is analogous to the conventional MCTS algorithm (Sutton & Barto, 2018), but differs in that it excludes fully-explored subtrees; refer to Algorithm 1 and its description.

These search strategies are employed to analyze the abilities of Transformers to navigate structured yet unknown environments. The pseudocode of these algorithms is shown in Section B. For the sake of brevity, uniform leaf sampling and greedy leaf sampling are categorized as *leaf-based sampling*, and uniform path sampling and policy-guided path sampling are categorized as *path-based sampling*.

### 3.3 Model Interfaces to Perform Search

Our problem setup can be viewed as a multi-turn interaction between an agent and an environment: At each step, the agent sends the environment a message containing the node $s$ it wants to visit, and the environment replies with the node's value $v$ and neighbors $N(s)$ (or unvisited states $F$).

This formulation allows Transformers or potentially any autoregressive next-token prediction model to serve as search policies by predicting the agent's messages in the conversation conditioned on the chat history. We analyze this search capability through three complementary lenses:

- Theoretically demonstrating the existence of Transformers' parameters that implement search algorithms (see Section 4);

- Training Transformers on algorithmic traces (see Section 5);

- Improving the performance of pretrained LLMs through targeted fine-tuning (see Section 6).

Since our tokenization schemes are essential components of both the theoretical and empirical analyses, we specify them in the respective sections and Section C, which provides detailed implementation information. Moreover, our theoretical analysis examines how Transformers' parameters represent search strategies, while our empirical analysis investigates their optimizability and generalizability. Together, these findings enhance our comprehensive understanding of Transformers' representational and performance capabilities in search.

## 4 Theoretical Analysis of Transformers' Search Capability

In this section, we identify the theoretical search capabilities of LLMs by demonstrating that Transformers (Vaswani et al., 2017) can implement the search algorithms introduced in Section 3 by simulating each step of the algorithms. For any given trajectory of past search steps, each search strategy will define a probability distribution over the next state to visit. We thus show that when the trajectory is provided as input (encoded as a sequence of tokens), the Transformer can output a distribution over next states that exactly matches the distribution prescribed by the target search strategy. To do this, we provide explicit weight constructions for both leaf-based and path-based sampling methods, using Transformers with constant depth and embedding dimension linear in search budget $T$ and branching factor $B$. Moreover, in our theoretical analysis, a specific Transformer model is assumed: (i) layer normalization is omitted; (ii) the fully connected layer is replaced with any arbitrary token-wise function; (iii) the conventional softmax attention mechanism is replaced with hard attention. This specific model differs from the actual implementation of Transformers for empirical analysis. See Section D.1 for the details of our theoretical Transformer model.

Given a search step budget $T$ and a maximum branching factor $B$, the trace of the search trajectory can contain at most $TB + 1$ distinct states as there is also the root node. For simplicity, we assume exactly $TB + 1$ states in total. Then, we define two tokenization schemes, *Leaf-Based Tokenization* and *Tree-Based Tokenization*.

**Leaf-Based Tokenization**   We define three token types for leaf-based search methods:

- State tokens $S_i$ for each unique state $i$;

- Continuous value tokens $V_\alpha$, where $\alpha \in [0, 1]$ is stored in a dedicated coordinate in the embedding;

- Structural markers %, #, and ?.

Let a trajectory be
$$\tau = ((s_0, v_0, N(s_0)), (s_1, v_1, N(s_1)), \ldots, (s_T, v_T, N(s_T))), \tag{7}$$

where $N(s_t) = \{s_{t,1}, s_{t,2}, \ldots\}$ is the set of $s_t$'s child states. We then construct the tokenized trace as follows:

$$? \; S_{s_0} \; \% \; V_{v_0} \; \# \; S_{s_{0,1}} \; S_{s_{0,2}} \; \ldots \; ? \; S_{s_1} \; \% \; V_{v_1} \; \# \; S_{s_{1,1}} \; S_{s_{1,2}} \; \ldots \; ? \; \ldots$$

for $t = 0, 1, \ldots, T$. Here, the following special tokens are defined:

- % separates the selected state from its value embedding.

- # precedes the list of child states.

- ? terminates each search iteration.

Note that % is not required for the leaf-based sampling theoretical construction and it is added to ensure consistency with the tree-based sampling methods below. We will construct a Transformer model to predict the next state token $S_{s_t}$ immediately after each ?.

**Tree-Based Tokenization** For path-based search algorithms, we extend the leaf-based trace with the start-of-sequence marker `[BOS]` and the path separator `>`. At each iteration, we encode the full tree-policy path leading to the chosen state. The resulting sequence is as follows:

$$\texttt{[BOS]} \; ? \; \pi_{\text{tree}}^{(0)} \; \texttt{\%} \; V_{v_0} \; \texttt{\#} \; S_{s_{0,1}} \; S_{s_{0,2}} \; \ldots \; ? \; \pi_{\text{tree}}^{(1)} \; \texttt{\%} \; V_{v_1} \; \texttt{\#} \; S_{s_{1,1}} \; S_{s_{1,2}} \; \ldots \; ? \; \ldots$$

where the following:

$$\pi_{\text{tree}}^{(t)} = S_{s_0} \; \texttt{>} \; S_{s_{0,i_1}} \; \texttt{>} \; \cdots \; \texttt{>} \; S_{s_t}, \tag{8}$$

denotes the sequence of states selected by the tree policy, whose terminal state $S_{s_t}$ is chosen for expansion at step $t$. Our constructed model will generate the complete path $\pi_{\text{tree}}^{(t)}$ at each step, conditioned on the previous trace. The Transformer terminates the generation by outputting the `%` token, which also indicates that the immediately preceding state token is the state that should be used for expansion in the next step.

More details of these tokenization schemes are described in Section C.1.

We now present the following theorems:

**Theorem 1 (Leaf-Based Search Policies)** *There exist 3-layer Transformers with embedding dimension $d = 10 + TB$ that exactly implement the uniform and greedy leaf sampling policies when using a sequential encoding of the trajectory using* Leaf-Based Tokenization.

**Theorem 2 (Path-Based Search Policies)** *There exist 12-layer Transformers with embedding dimension $d = 26 + 7TB$ that exactly implement the uniform path sampling policy and greedy, pure exploration and UCT based path sampling policies when using a sequential encoding of the trajectory using* Tree-Based Tokenization.

The full proofs of Theorems 1 and 2 are provided in Appendices D.2 and D.3, respectively. Here, we present a high-level overview of the proof for Theorem 1, specifically on how to construct a Transformer that implements the greedy leaf policy. The construction proceeds in three layers:

(i) Token marking: The first layer uses the input embeddings as a lookup table to identify token types (e.g., state tokens). It then marks whether each state token was *selected* by the model (a visited node) or *given* by the environment (a child node of a visited node).

(ii) Frontier identification and value association: The second layer uses the markings from the previous layer to isolate the frontier states (the unvisited children of visited nodes). For each frontier state, it identifies and retrieves the rollout value from its parent node.

(iii) Selection: The third layer applies a maximum operation over the inherited rollout values associated with the frontier states to select the next state to visit.

The proof for Theorem 2 is similar in flavor, using the same proof techniques. The additional complexities arise from needing to first simulate the path policy before doing selection.

As a final note on the theory, we observe that the leaf-based tokenization is linear in the number of iterations, while the tree-based one is at worst quadratic. Thus both are polynomial in the number of search iterations. We also emphasize that the theoretical significance of these two theorems lies in providing guarantees with the following properties: (i) the Transformer constructions have constant depth; (ii) the guarantees are in the form of exact implementations. These results offer a stronger guarantee than existing universal approximation theorems (e.g., the work by Yun et al. (2020)), which only ensure approximate implementation over compact domains and require non-constant depth.

> **Finding 1**: Transformers possess a sufficient architectural capacity to implement fundamental search algorithms including leaf-based and path-based search policies.

Along with the above finding, one caveat however merits emphasis: While these existence results establish architectural feasibility, they do not address whether such implementations can be learned through standard

training paradigms. Specifically, the Transformers' parameters proved in Theorems 1 and 2 can be thought of as ideal parameters. The practical ability of Transformers to implement these search algorithms depends on whether they can learn the correct parameterizations solely from traces of the reference algorithms, which we empirically explore in the following section.

## 5 Empirical Analysis on Transformers' Search Abilities

Following the theoretical analysis presented in Section 4, we empirically test Transformers' behavior cloning abilities. We begin by introducing the additional metrics we use for evaluation.

**Experimental Details**  The architecture of Llama models (Touvron et al., 2023; Llama Team, AI @ Meta, 2024) is adopted to our Transformer models that are trained from scratch. Unless otherwise specified, the Transformer models are defined with 8 blocks, 8 heads, and 512 hidden dimensions. The number of intermediate dimensions is set as 1,024. Minimum and maximum learning rates are $5 \times 10^{-5}$ and $5 \times 10^{-4}$. It uses a learning rate scheduling technique with a warm-up. The AdamW optimizer (Loshchilov & Hutter, 2018) with $\beta_1 = 0.9$ and $\beta_2 = 0.99$ is utilized, and the rotary positional embedding (Su et al., 2024) with $\theta = 10000$ is used.

As offline training and validation datasets, we generate 200 different problem instances and 100 traces per instance, for a total of 20,000 traces. These datasets are split into 70% training and 30% validation. On the other hand, we report online test performance for all experiments. These test experiments use 10 different problem instances and 100 traces per instance, for a total of 1,000 traces. These results with 1,000 traces are used to calculate the mean and standard deviation of each experiment set. We verify that training and test problem instances are mutually exclusive. Moreover, we plot 1.96 standard errors for all figures.

**Evaluation Metrics**  We use the following evaluation metrics to analyze search algorithms:

- Max-reward hit rate: The fraction of runs that ever attain the global-best reward $r(s_{\text{best}})$ calculates the probability of achieving $r(s_{\text{best}})$ over multiple runs (i.e., the hit rate): $N_{\text{hit}}/N_{\text{total}}$, where $N_{\text{hit}}$ and $N_{\text{total}}$ are the number of runs that hit $s_{\text{best}}$ and the total number of runs, respectively;

- Discounted cumulative gain: This metric, based on discounted cumulative gain (Järvelin & Kekäläinen, 2002), quantifies how quickly a search algorithm finds $s_{\text{best}}$ in each run: $(1/N_{\text{total}}) \sum_{k=1}^{N_{\text{total}}} 1/\log_2(i_k + 1)$, where $i_k$ is the 1-indexed max-reward hit iteration index at run $k$. If a particular run fails, $i_k = \infty$;

- Normalized path length: This metric regarding the shortest path to $s_{\text{best}}$ computes a metric value of $(1/N_{\text{total}}) \sum_{k=1}^{N_{\text{total}}} \exp(\ell_{k,\text{truth}} - \ell_k)$, where $\ell_{k,\text{truth}}$ is the ground-truth shortest path length to $s_{\text{best}}$ and $\ell_k$ is the shortest path length predicted by the Transformer model. If it fails to find $s_{\text{best}}$, $\ell_k = \infty$, which makes $\exp(\ell_{k,\text{truth}} - \ell_k)$ zero;

- Highest/cumulative reward: The highest reward is calculated by $(1/N_{\text{total}}) \sum_{k=1}^{N_{\text{total}}} \max(\mathcal{R}_k)$ and a cumulative reward is calculated by $(1/N_{\text{total}}) \sum_{k=1}^{N_{\text{total}}} \sum_{r \in \mathcal{R}_k} r$, where $\mathcal{R}_k$ is the rewards achieved at run $k$. Compared to the max-reward hit rate, these account for both suboptimal and global-best rewards.

- Normalized jump distance: This is the arithmetic mean of jump distances where a jump distance is defined by $(\text{jump}(s_{t-1}, s_t) + \text{jump}(s_t, s_{t+1}))/2$, which has been proposed in the work by Zeng et al. (2025). Note that $\text{jump}(s_i, s_j)$ indicates the shortest distance between $s_i$ and $s_j$ on a tree-structured space.

These metrics are selected to evaluate search algorithms across criteria such as efficiency, robustness, and solution quality. Note that higher values indicate better performance across all metrics except for normalized jump distance.

**Transformers' Behavior Cloning Abilities**  We show that Transformers trained from scratch are capable of effectively performing behavior cloning in the two environments presented in Section 3.1. Full details of the tokenization method are in Section C.2.

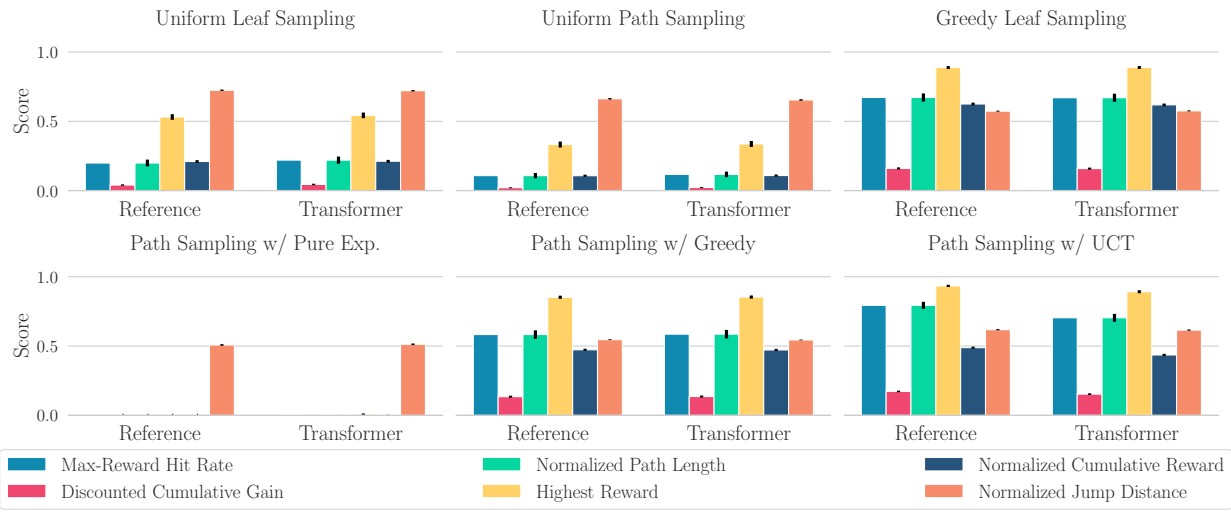

Figure 3: Behavior cloning results on the multi-reward tree search problem, where each binary tree of depth 6 has 8 different goals and a search step budget is 50.

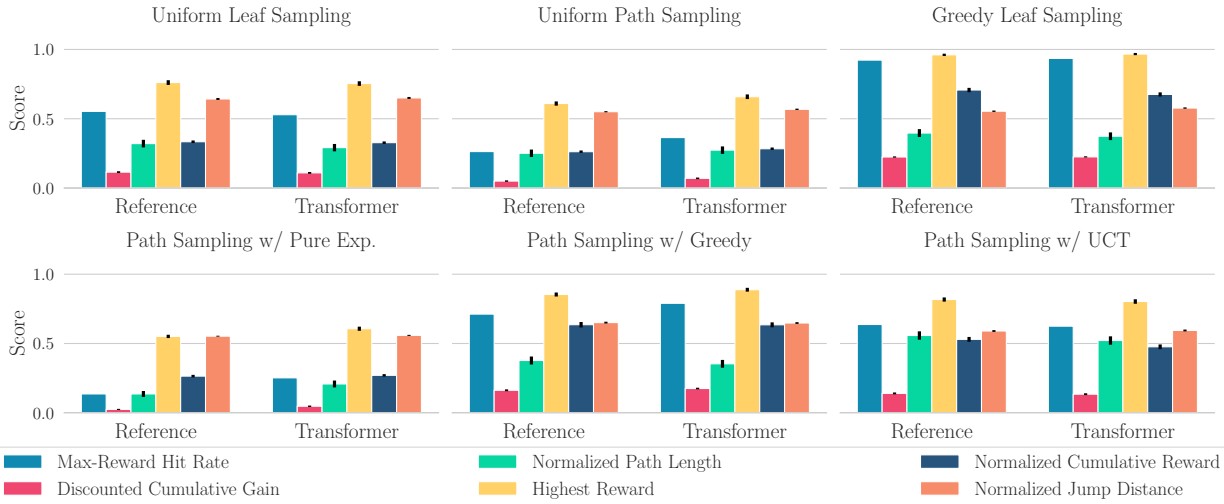

Figure 4: Behavior cloning results on the multi-reward navigation problem, where the size of each problem is $4 \times 4$, the wall density of each problem is 0.4, and a search step budget is 50.

As shown in Figures 3 and 4, Transformers successfully mimic the performance of the respective reference algorithms. To quantitatively analyze these results, we calculate $\ell^2$ distance between two vectors of the statistics of the metric values obtained by a reference algorithm and its corresponding Transformer model. We simply employ their means and standard deviations of all six metric values to measure the distance. Figure 5 displays that the results of Transformers are mostly closest to ones of the associated reference algorithms.

Beyond matching the reference algorithm's performance, we test whether the Transformer replicates its decision process by measuring the KL divergence between their next-state selection distributions under the same trajectory. This KL divergence is computed online by following the traces of the reference algorithms and comparing the predicted next-state probabilities generated by the reference strategies and the Transformer models. Eventually, these probabilities are used for calculating the KL divergence $D_{\mathrm{KL}}$:

$$D_{\mathrm{KL}} = \sum p(x) \log \left( \frac{p(x)}{q(x)} \right), \tag{9}$$

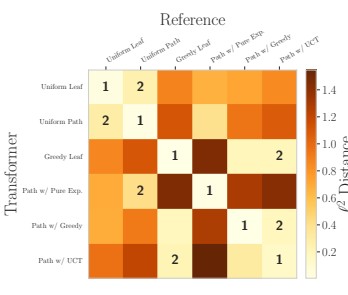

(a) Multi-reward tree search

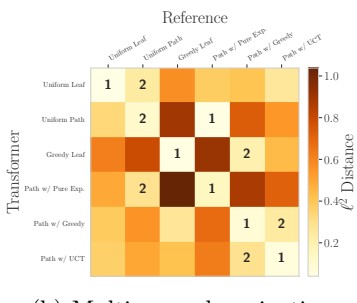

(b) Multi-reward navigation

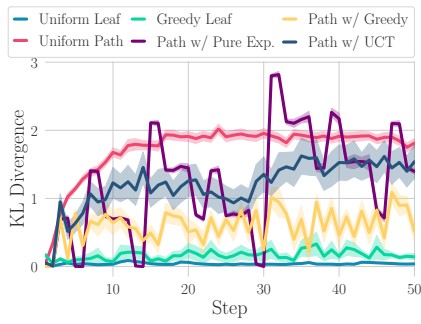

Figure 5: Comparison of the metric values obtained by reference algorithms and Transformers using the results shown in Figures 3 and 4. $\ell^2$ distance between two vectors regarding the metrics is presented. The shortest and second shortest distances in each row are marked as 1 and 2, respectively.

Figure 6: KL divergences between the probabilities of reference algorithms and Transformers. The traces of reference algorithms are followed for both.

where $p(x)$ and $q(x)$ is the next-token probabilities over a token $x$ of a reference algorithm and Transformer model, respectively. Note that the next-state probabilities of the reference algorithms reflect uniform tie-breaking among equally preferred nodes. In particular, to calculate the KL divergence, we should obtain the next-state probabilities of reference algorithms. Uniform leaf sampling assigns equal probabilities to all possible next states, and uniform path sampling uses depth-level equal probabilities to calculate the probabilities of possible next states. Greedy and policy-guided path sampling follow their respective tree traversal rules, while also considering all tied states when selecting among equally preferred options. As a result, we empirically estimate the probabilities of the reference algorithms by repeating the sampling process 100 times with the same trace, varying only the random seed.

As shown in Figure 6, the leaf sampling algorithms achieve low KL divergence with their reference counterparts, suggesting successful behavior alignment. In contrast, for path sampling algorithms, they exhibit significantly higher KL divergence, compared to the leaf sampling algorithms. This suggests that they are unlikely to fully replicate the reference algorithms' behavior despite matching the performance on all of our metrics; the KL divergence results do not agree with the resulting performance reported in Figure 5. This discrepancy arises because of the following rationales: (i) the training traces for path-based methods do not explicitly encode the tree-policy path for each step; (ii) the randomness included in the process of particular path-based methods, such as the uniform path sampling, pure exploration policy-guided path sampling, and UCT policy-guided path sampling, makes KL divergence higher. Regarding the first rationale, the Transformer must infer the tree-based policy and select the next state in a single step, without intermediate tree traversal-related outputs. Due to this, it is a naturally challenging problem under our tokenization scheme. In addition, this aligns with our theoretical analysis in Section 4, where the trace format as described in Section C.1 explicitly requires the Transformer to output the tree traversal policy before selecting the next state. Regarding the second rationale, the order of some subsequent states may be interchangeable, since their ordering does not significantly affect the eventual outcome. This rationale is supported by two observations: (i) pure exploration policy-guided path sampling shows periodic KL divergence patterns with recurring low-divergence regions; (ii) greedy policy-guided path sampling yields lower KL divergence than other policy-guided path sampling methods.

We carry out generalization analysis on the numbers of test steps, test wall densities, the numbers of test goal states, and test tree depths in Figures 7, 8, 19, and 20, respectively. These empirical results show that our behavior-cloned Transformers present evidence of generalization to novel tasks. In particular, our Transformer models generalize well within a local range, whereas their generalization performance degrades over longer ranges. We conjecture that this behavior stems from practical limitations such as short context length and limited model capacity. It is noteworthy that we test unseen tasks that are harder than training tasks, similar to the recent work by Lee et al. (2025). For example, later test steps are more difficult than earlier ones, as the search space expands with each step.

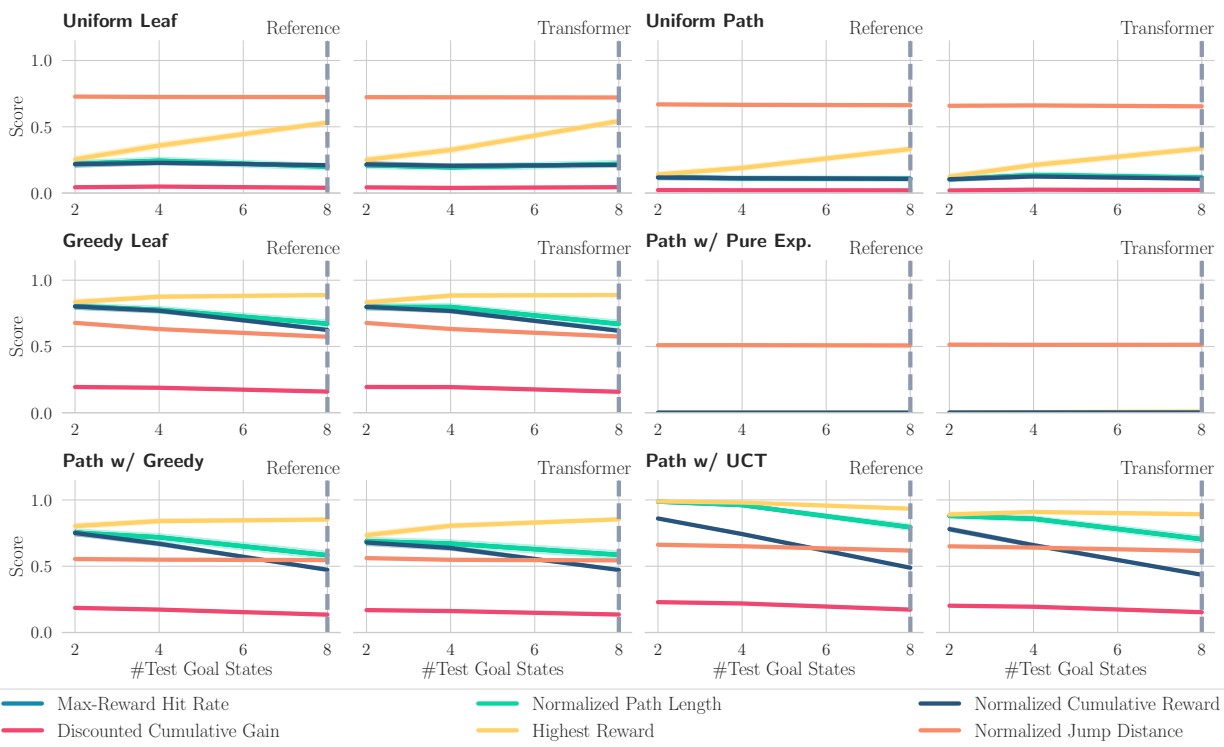

Figure 7: Generalization analysis over the numbers of test goal states on the binary tree search problem of depth 6 with a step budget of 50. The number of test goal states smaller than 8 is unseen. A gray dashed line indicates the setting used in training.

Table 1: Comparison of Qwen3-8B and fine-tuned Qwen3-8B models, with and without UCT policy-guided sampling. FT stands for fine-tuned.

| Performance Metric | Qwen3-8B | Qwen3-8B + UCT Policy | FT Qwen3-8B | FT Qwen3-8B + UCT Policy |
|---|---|---|---|---|
| Max-Reward Hit Rate, Highest Reward, & Cumulative Reward | 0.262 (0.069) | 0.286 (0.071) | 0.500 (0.078) | 0.548 (0.078) |
| Discounted Cumulative Gain | 0.153 (0.046) | 0.188 (0.052) | 0.262 (0.050) | 0.285 (0.047) |
| Normalized Path Length | 0.262 (0.069) | 0.286 (0.071) | 0.449 (0.075) | 0.497 (0.076) |

> **Finding 2**: The Transformers trained from scratch are capable of approximating diverse search algorithms via behavior cloning and exhibiting the possibility of generalizing to unseen tasks.

More results and discussion on Transformers' behavior cloning abilities can be found in Section F.1.

## 6  Targeted Fine-Tuning for Enhancing Search Capabilities of LLMs

There is room for improving the search abilities of existing LLMs by utilizing targeted fine-tuning. To tackle this issue, we train a pretrained LLM, i.e., Qwen3-8B (Qwen Team, 2025), on the search trajectories of prompts and their associated responses for the Academic Paper Search problem, where this problem requires an agent to navigate from a start instance to a target by interacting with the corresponding environments. We choose this problem because its state expansion and evaluation are more structured than in more generic reasoning tasks such as GSM8K (Cobbe et al., 2021) and HotpotQA (Yang et al., 2018). The Academic Paper Search problem provides a fixed set of state expansions, since each paper has a predetermined set of child papers, and supports semantic-based state evaluation, since the distance between two papers can be

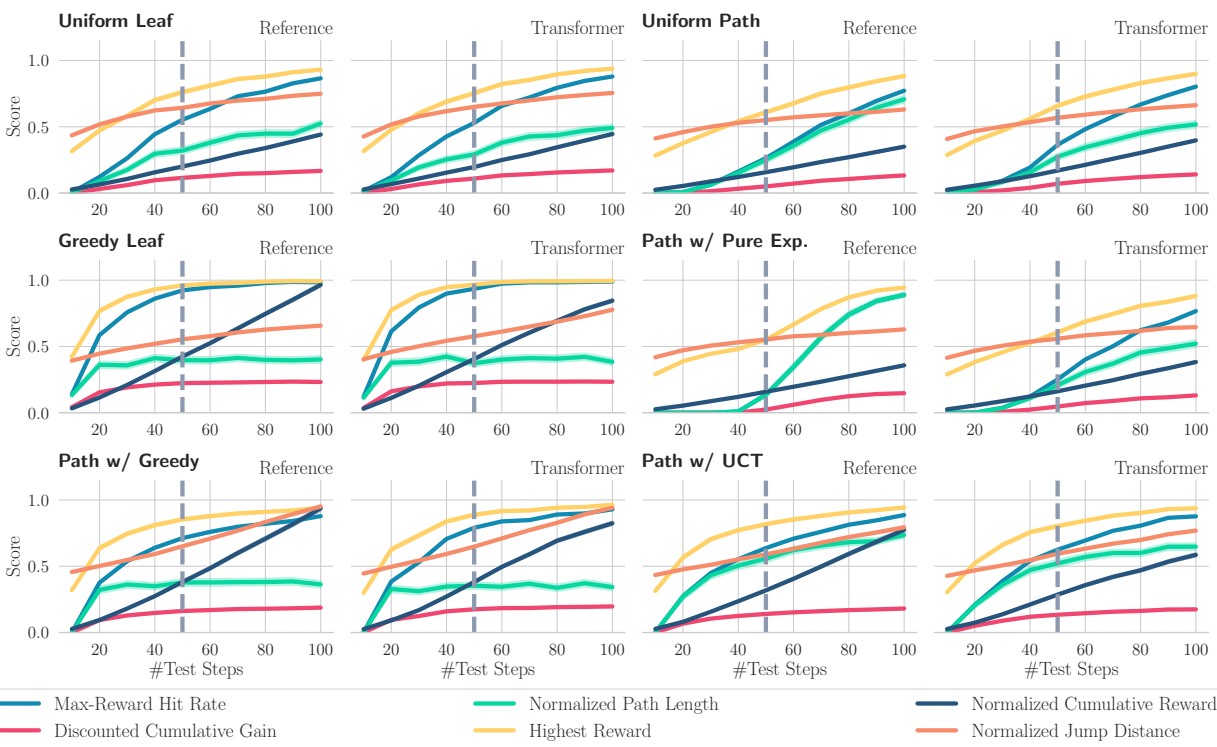

Figure 8: Generalization analysis over the numbers of test steps on the multi-reward navigation problem of size $4 \times 4$ with a wall density of 0.4. The number of test steps larger than 50 is unseen in a training phase. Transformer-based models adequately follow their reference algorithms in both in-distribution and out-of-distribution settings. A gray dashed line indicates the setting used in training.

defined in a straightforward manner. Consequently, this setting aligns well with our problem formulation of unknown tree search with bandit feedback.

The trajectories generated by Gemini-2.5-Flash and policy-guided path sampling with UCT are used for training the pretrained LLM. To verify the performance of a fine-tuned LLM, we compare the fine-tuned LLM to two methods: (i) a pretrained LLM; (ii) the pretrained LLM with a specialized search algorithm. Specifically, the pretrained and fine-tuned LLMs perform all three steps of the effective problem-solving process, i.e., space expansion, next-state selection, and state evaluation, whereas the pretrained LLM combined with a search algorithm carries out only the two steps, space expansion and state evaluation, with the external algorithm handling next-state selection. The missing details of this experiment can be found in Section E.2.

As shown in Table 1, our fine-tuned Qwen3-8B is significantly superior to the two baseline methods in this problem. These results imply that the search abilities of LLMs are not fully exploited and the additional training explicitly designed for search can successfully unlock these abilities. Importantly, it aligns with our findings discussed in Sections 4 and 5, which demonstrate that the Transformer models, i.e., the underlying architectures of LLMs, are capable of implementing various search algorithms both theoretically and empirically.

> **Finding 3**: Targeted fine-tuning enhances the search capabilities of LLMs compared to both an LLM-only method and an LLM with an external algorithm.

Despite our successful experimental results presented in this section, the LLM used in these settings may introduce unknown biases into the trained model. This issue can potentially be mitigated by using the search trajectories obtained from multiple LLMs. In addition, fine-tuned Qwen3-8B with the specialized

search algorithm is the best compared to the other methods. It is a natural outcome since performance through an LLM with an external search algorithm can be considered as ground-truth performance. By optimizing a fine-tuned model with a search algorithm, the model's performance could be further improved; this exploration is left for future work.

## 7 Discussion and Conclusion

To understand whether LLMs truly perform structured search or not, we have tackled the search problem itself, where space extensions and bandit feedback are externally given. Then, we show both theoretically and empirically, that Transformers can approximately implement diverse search algorithms. In addition to these theoretical and empirical results, we empirically demonstrate that the trained Transformers showcase evidence of generalization to unseen problem conditions. Finally, the additional fine-tuning of an existing LLM allows us to boost its search abilities.

**Limitations and Future Directions**  Since multi-turn interactions between an agent and an environment are assumed, the process of determining a final state or an answer tends to be inefficient. If this overall search process can be internalized within LLMs (Deng et al., 2024; Hao et al., 2024), it may lead to a more efficient alternative to the current approach. Furthermore, to scale up our experiments, we should investigate the search capabilities of Transformers on significantly longer sequences within larger problem instances, which would require larger model sizes and greater computational resources (Kaplan et al., 2020). The search behavior of Transformers in those settings may differ from the current observations. Although this work empirically demonstrates the optimization and generalization of Transformers' parameters for implementing search algorithms, it does not address them theoretically; future work can extend it to establish their theoretical learnability and generalizability by following existing work (Zhang et al., 2024). While our empirical analysis demonstrates the effectiveness of Transformer models in terms of generalization, their generalization abilities still remain limited. Therefore, as future work, one possible direction is to explore a more extensive analysis of generalization, similar to existing studies (Lee et al., 2025; Golowich et al., 2025; Cai et al., 2025). On the other hand, we anticipate that RL can enhance the performance of Transformer-based search algorithms with respect to a given evaluation metric following recent literature in RL for LLMs (DeepSeek-AI, 2025; Shenfeld et al., 2025). Our framework for enhancing the search capabilities of LLMs has the potential to address real-world problems such as planning-based prompting on trees by leveraging tools introduced in recent literature (Xiong et al., 2025; Lu et al., 2025; Minegishi et al., 2025); we leave this for future work. These applications will enable us to investigate the search performance of LLMs from a practical standpoint. Finally, as discussed in Section 6, while targeted fine-tuning helps internalize the search capabilities of LLMs, it is still hard to reach the performance through exact search algorithms. Following previous work on tool use (Yao et al., 2023b; Schick et al., 2023), it implies that the use of external search algorithms may be a more practical strategy, even if this approach does not internalize the search capabilities within the LLM.

### Broader Impact Statement

This work is not directly related to any unethical or harmful tasks since it focuses on the understanding of Transformer models and potentially the improvement of LLMs. However, similar to many advances in learning algorithms, there is a risk that improved search capabilities could be applied to harmful tasks, such as optimizing for undesired behaviors in interactive systems. Therefore, we should mind this risk and ensure alignment with ethical standards.

### Acknowledgments

This work was supported by National Science Foundation (NSF) award DMS-2023239, NSF CAREER award CCF-2339978, the Amazon Research Award, the Google Cloud Research Credits Program, and a grant from FuriosaAI. In addition, it used resources through CIS251382 from the Advanced Cyberinfrastructure Coordination Ecosystem: Services & Support (ACCESS) program, which is supported by NSF awards OAC-2138259, OAC-2138286, OAC-2138307, OAC-2137603, and OAC-2138296, and resources from the University

of Pittsburgh Center for Research Computing and Data (RRID:SCR_022735), supported by National Institutes of Health (NIH) award S10OD028483 and NSF award OAC-2117681. We also acknowledge partial support from the National Research Foundation of Korea (NRF) grant funded by the Ministry of Science and ICT (MSIT), Korea (RS-2024-00345351, RS-2024-00408003); the ICT Challenge and Advanced Network of HRD (ICAN) Support Program (RS-2023-00259934, RS-2025-02283048) supervised by the Institute for Information & Communications Technology Planning & Evaluation (IITP); and the Yonsei University Research Fund (2025-22-0025).

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

# A    Construction of Tree-Structured Spaces for Multi-Reward Navigation

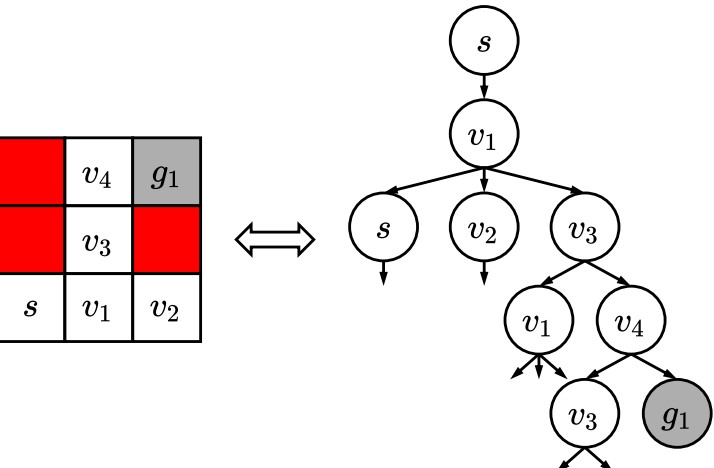

Figure 9: Construction of a tree-structured space from a particular instance of multi-reward navigation. $s$ and $g_1$ are a start node and a goal node, respectively. In addition, $v_1$, $v_2$, $v_3$, and $v_4$ are visitable nodes.

Figure 9 presents how a tree-structured search space is constructed from a particular instance of multi-reward navigation. Deeper nodes are omitted for clarity.

# B   Details and Pseudocode of Reference Search Algorithms

In this section, we provide the pseudocode of reference search algorithms.

---

**Algorithm 1** Fully-Explored Sub-Tree Exclusion

---

**Input:** The current state $s$, unvisited child states $F$, a hidden search tree $\mathcal{T} = (\mathcal{S}, N)$.
**Output:**
1: Obtain all visitable child states of $s$, denoted as $\mathcal{C}$.
2: **if** $\mathcal{C} \cap F = \emptyset$ **then**
3:    **return** True
4: **else**
5:    **return** False
6: **end if**

---

Algorithm 1 verifies if all possible child states of a state $s$ are all explored. This algorithm is denoted as $E(s \mid F, \mathcal{T})$, where $F$ is unvisited child states and $\mathcal{T}$ is a hidden search tree. If $E(s \mid F, \mathcal{T})$ is True, a search algorithm should exclude this sub-tree of which the root is $s$ from further search. Importantly, with this process, underlying tree structures are not given to search algorithms, and we only provide unvisited states to the algorithms.

Without loss of generality, we can define a modified successor function $N^*$, which only provides immediate child states that have unvisited states:

$$N^*(s) = \{s' \in N(s) \mid \neg E(s' \mid F, \mathcal{T})\}, \tag{10}$$

where $\neg$ is a logical not operation.

## B.1   Uniform Leaf Sampling

---

**Algorithm 2** Uniform Leaf Sampling

---

**Input:** A root state $s_0$, a value estimation function $\widehat{V}(s)$, a step budget $T$, a hidden search tree $\mathcal{T} = (\mathcal{S}, N)$.
**Output:** A search trajectory $\tau = [(s_0, v_0, N(s_0)), (s_1, v_1, N(s_1)), \ldots, (s_T, v_T, N(s_T))]$.
1: Initialize a trajectory $\tau = [(s_0, v_0, N(s_0))]$.
2: Update unvisited child states $F_0$.
3: **for** $t = 1, 2, \ldots, T$ **do**
4:    Choose the next state $s_t$ from $F_{t-1}$ uniformly at random.
5:    Evaluate $s_t$ to obtain its value $v_t$ using $\widehat{V}(s_t)$.
6:    Update $\tau$ with $(s_t, v_t, \widehat{V}(s_t))$.
7:    Update unvisited child states $F_t$.
8: **end for**

---

Algorithm 2 shows the pseudocode of uniform leaf sampling. This algorithm can be considered as the randomized algorithm of DFS.

## B.2 Greedy Leaf Sampling

---

**Algorithm 3** Greedy Leaf Sampling

---

**Input:** A root state $s_0$, a value estimation function $\widehat{V}(s)$, a step budget $T$, a hidden search tree $\mathcal{T} = (\mathcal{S}, N)$.
**Output:** A search trajectory $\tau = [(s_0, v_0, N(s_0)), (s_1, v_1, N(s_1)), \ldots, (s_T, v_T, N(s_T))]$.
 1: Initialize a trajectory $\tau = [(s_0, v_0, N(s_0))]$.
 2: Update unvisited child states $F_0$.
 3: **for** $t = 1, 2, \ldots, T$ **do**
 4:     Choose a state $s'_t$ from the parent states of $F_{t-1}$ which has the highest reward estimated value.
 5:     Sample the next state $s_t$ from child states $N(s'_t)$ uniformly.
 6:     Evaluate $s_t$ to obtain its value $v_t$ using $\widehat{V}(s_t)$.
 7:     Update $\tau$ with $(s_t, v_t, \widehat{V}(s_t))$.
 8:     Update unvisited child states $F_t$.
 9: **end for**

---

Algorithm 3 demonstrates the pseudocode of greedy leaf sampling. It considers the reward values of the parent states of unvisited states.

## B.3 Uniform Path Sampling

---

**Algorithm 4** Uniform Path Sampling

---

**Input:** A root state $s_0$, a value estimation function $\widehat{V}(s)$, a step budget $T$, a hidden search tree $\mathcal{T} = (\mathcal{S}, N)$.
**Output:** A search trajectory $\tau = [(s_0, v_0, N(s_0)), (s_1, v_1, N(s_1)), \ldots, (s_T, v_T, N(s_T))]$.
 1: Initialize a trajectory $\tau = [(s_0, v_0, N(s_0))]$.
 2: Update unvisited child states $F_0$.
 3: **for** $t = 1, 2, \ldots, T$ **do**
 4:     Assign a state as $s'_t = s_0$.
 5:     **while** $s'_t \notin F_{t-1}$ **do**
 6:         Choose one of $N^*(s'_t)$ uniformly at random, ensuring that fully-explored sub-trees are not considered.
 7:         Update $s'_t$ using the sampled state.
 8:     **end while**
 9:     Assign the next state as $s_t = s'_t$
10:     Evaluate $s_t$ to obtain its value $v_t$ using $\widehat{V}(s_t)$.
11:     Update $\tau$ with $(s_t, v_t, \widehat{V}(s_t))$.
12:     Update unvisited child states $F_t$.
13: **end for**

---

Algorithm 4 presents a search algorithm of uniform path sampling. In contrast to uniform leaf sampling, this algorithm can be viewed as a randomized variant of BFS. The definition of $N^*$ is shown in (10).

### B.4 Path Sampling with Pure Exploration, Greedy, or UCT Policy

---

**Algorithm 5** Path Sampling with a Specific Tree Traversal Policy

---

**Input:** A root state $s_0$, a value estimation function $\widehat{V}(s)$, a tree traversal policy $\mathcal{P}$, a step budget $T$, a hidden search tree $\mathcal{T} = (\mathcal{S}, N)$.
**Output:** A search trajectory $\tau = [(s_0, v_0, N(s_0)), (s_1, v_1, N(s_1)), \ldots, (s_T, v_T, N(s_T))]$.
 1: Initialize a trajectory $\tau = [(s_0, v_0, N(s_0))]$.
 2: Update unvisited child states $F_0$.
 3: **for** $t = 1, 2, \ldots, T$ **do**
 4: Assign a state as $s'_t = s_0$.
 5: **while** $s'_t \notin F_{t-1}$ **do**
 6:  Determine unvisited immediate child states $\widetilde{F}_{t-1} = \{s : s \in N(s_t), s \in F_{t-1}\}$.
 7:  **if** $|\widetilde{F}_{t-1}| > 0$ **then**
 8:   Choose one of $\widetilde{F}_{t-1}$ uniformly at random.
 9:  **else**
10:   Choose one of $N^*(s'_t)$ with a specific tree traversal policy $T$; see Algorithm 6, ensuring that fully-explored sub-trees are not considered.
11:  **end if**
12:  Update $s'_t$ using the chosen state.
13: **end while**
14: Assign the next state as $s_t = s'_t$
15: Evaluate $s_t$ to obtain its value $v_t$ using $\widehat{V}(s_t)$.
16: Update $\tau$ with $(s_t, v_t, \widehat{V}(s_t))$.
17: Update unvisited child states $F_t$.
18: **end for**

---

Algorithm 5 shows a search strategy of path sampling with a specific tree traversal policy such as a pure exploration, greedy, or UCT policy. $N^*$ is defined in (10).

---

**Algorithm 6** Tree Traversal Policies

---

**Input:** A tree traversal policy $\mathcal{P}$, the current state $s$, modified successor function $N^*$.
**Output:** A selected next state $s^*$.
 1: **Pure Exploration Policy**
 2: $s^* = \arg\min_{s' \in N^*(s)} \text{count}(s')$, where count returns the number of visit counts.
 3: **Greedy Policy**
 4: $s^* = \arg\max_{s' \in N^*(s)} \text{value}(s')$, where value returns the summation of all values of the child nodes of $s'$.
 5: **UCT Policy**
 6: $s^* = \arg\max_{s' \in N^*(s)} \text{value}(s')/\text{count}(s') + c\sqrt{\log(2\,\text{count}(s))/\text{count}(s')}$, where $c$ is a balancing hyperparameter.

---

Algorithm 6 presents the details of tree traversal policies. The pure exploration policy only considers the number of visit counts and chooses the least visited state as the next state. The greedy policy takes into account the values of child states in order to choose the next state. The UCT policy (Kocsis & Szepesvári, 2006) balances exploration and exploitation considering both the number of visits and the estimated values. Across all experiments, we use $c = 0.1$. See this reference (Sutton & Barto, 2018) for more details.

## C  Trace Formats

This section presents the trace formats used for theoretical constructions, Transformers trained from scratch, and pretrained existing LLMs.

### C.1  Trace Format for Theoretical Constructions

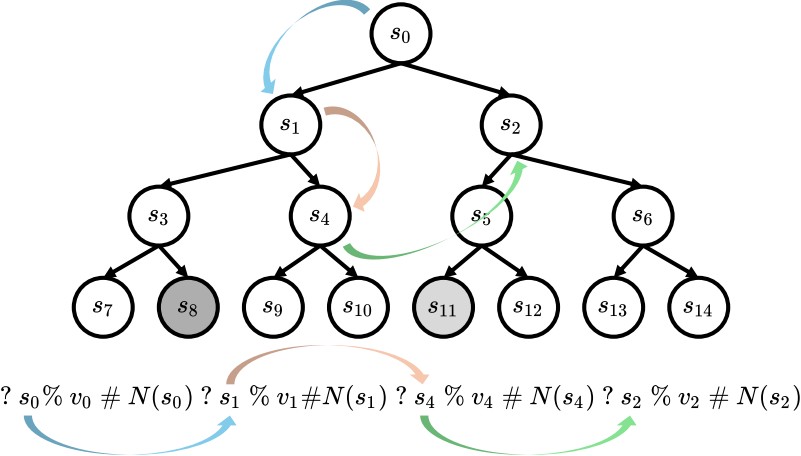

Figure 10: Visualization of our tokenization scheme. Same colors indicate same state transitions.

In our theoretical analysis, we represent each search trace as a sequence of discrete tokens. Figure 10 shows the tokenization scheme for leaf sampling algorithms.

### C.2  Trace Format for Transformers Trained from Scratch

To simplify both training and inference of Transformer models trained from scratch, we provide the full set of unvisited states $F_{t-1}$ at step $t$. Additionally, each unvisited state, denoted as $S_{t-1,1}, S_{t-1,2}, \ldots, S_{t-1,|F_{t-1}|}$, is accompanied by its 0-based index, denoted as $i_1, i_2, \ldots, i_{|F_{t-1}|}$. As a result, the Transformer models should predict these indices when predicting next states. It implies that the task becomes easier than the task of predicting states directly. The example of this trace format is as follows:

$$\cdots V_{t-1} \underbrace{\texttt{\#}\ i_1\ S_{t-1,1}\ i_2\ S_{t-1,2}\cdots i_{|F_{t-1}|}\ S_{t-1,|F_{t-1}|}\ \texttt{?}\ S_t\ V_t}_{\text{Tokens for a single step}}\ \texttt{\#}\ i_1\ S_{t,1}\ i_2\ S_{t,2}\cdots i_{|F_t|}\ S_{t,|F_t|}\ \cdots$$

where $V_t$ is an estimated value of $S_t$ at step $t$. In this setting, we discretize $V_t$ to two decimal places and represent it using 101 tokens corresponding to the values $0.00, 0.01, 0.02, ..., 0.99, 1.00$.

```
start_of_iteration 0 r0d0>i0d1 1 r0d0>i1d1 2 r0d0>i2d1 selected_child_and_then_reward
2 0.00 start_of_iteration 0 r0d0>i0d1 1 r0d0>i1d1 2 r0d0>i2d1>i0d2 3 r0d0>i2d1>i1d2 4
r0d0>i2d1>i2d2 selected_child_and_then_reward 0 0.00 start_of_iteration 0 r0d0>i1d1 1
r0d0>i2d1>i0d2 2 r0d0>i2d1>i1d2 3 r0d0>i2d1>i2d2 4 r0d0>i0d1>i0d2 5 r0d0>i0d1>i1d2 6
r0d0>i0d1>i2d2 selected_child_and_then_reward 6 0.00 start_of_iteration 0 r0d0>i1d1 1
r0d0>i2d1>i0d2 2 r0d0>i2d1>i1d2 3 r0d0>i2d1>i2d2 4 r0d0>i0d1>i0d2 5 r0d0>i0d1>i1d2 6
r0d0>i0d1>i2d2>i0d3 7 r0d0>i0d1>i2d2>i1d3 8 r0d0>i0d1>i2d2>i2d3 selected_child_and_then_reward
4 0.40 start_of_iteration 0 r0d0>i1d1 1 r0d0>i2d1>i0d2 2 r0d0>i2d1>i1d2 3 r0d0>i2d1>i2d2
4 r0d0>i0d1>i1d2 5 r0d0>i0d1>i2d2>i0d3 6 r0d0>i0d1>i2d2>i1d3 7 r0d0>i0d1>i2d2>i2d3 8
r0d0>i0d1>i0d2>i0d3 9 r0d0>i0d1>i0d2>i1d3 10 r0d0>i0d1>i0d2>i2d3 selected_child_and_then_reward
9 1.00 start_of_iteration 0 r0d0>i1d1 1 r0d0>i2d1>i0d2 2 r0d0>i2d1>i1d2 3 r0d0>i2d1>i2d2
4 r0d0>i0d1>i1d2 5 r0d0>i0d1>i2d2>i0d3 6 r0d0>i0d1>i2d2>i1d3 7 r0d0>i0d1>i2d2>i2d3 8
r0d0>i0d1>i0d2>i0d3 9 r0d0>i0d1>i0d2>i2d3 selected_child_and_then_reward 9 0.00
```

Figure 11: Trace example of multi-reward tree search problems.

```
start_of_iteration 0 x0y0>x0y1 1 x0y0>x1y0 selected_child_and_then_reward 0 0.00
start_of_iteration 0 x0y0>x1y0 1 x0y0>x0y1>x0y2 2 x0y0>x0y1>x0y0 3 x0y0>x0y1>x1y1
selected_child_and_then_reward 1 0.10 start_of_iteration 0 x0y0>x1y0 1 x0y0>x0y1>x0y0 2
x0y0>x0y1>x1y1 3 x0y0>x0y1>x0y2>x0y3 4 x0y0>x0y1>x0y2>x0y1 selected_child_and_then_reward 3
0.10 start_of_iteration 0 x0y0>x1y0 1 x0y0>x0y1>x0y0 2 x0y0>x0y1>x1y1 3 x0y0>x0y1>x0y2>x0y1
4 x0y0>x0y1>x0y2>x0y3>x0y4 5 x0y0>x0y1>x0y2>x0y3>x0y2 selected_child_and_then_reward 4 0.00
start_of_iteration 0 x0y0>x1y0 1 x0y0>x0y1>x0y0 2 x0y0>x0y1>x1y1 3 x0y0>x0y1>x0y2>x0y1 4
x0y0>x0y1>x0y2>x0y3>x0y2 5 x0y0>x0y1>x0y2>x0y3>x0y4>x0y3 6 x0y0>x0y1>x0y2>x0y3>x0y4>x1y4
selected_child_and_then_reward 4 0.00 start_of_iteration 0 x0y0>x1y0 1 x0y0>x0y1>x0y0
2 x0y0>x0y1>x1y1 3 x0y0>x0y1>x0y2>x0y1 4 x0y0>x0y1>x0y2>x0y3>x0y4>x0y3 5
x0y0>x0y1>x0y2>x0y3>x0y4>x1y4 6 x0y0>x0y1>x0y2>x0y3>x0y2>x0y3 7 x0y0>x0y1>x0y2>x0y3>x0y2>x0y1
selected_child_and_then_reward 3 0.00
```

Figure 12: Trace example of multi-reward navigation problems.

Figures 11 and 12 show the trace examples of multi-reward tree search and navigation problems. These tokens are split by whitespaces, which implies that a single word corresponds to a single token. For the multi-reward navigation problem, to reduce the number of state tokens, we additionally split the state tokens with ">."

### C.3 Trace Format for Pretrained Large Language Models

For the existing pretrained LLM, we first initialize it with the system prompt shown in Figure 13. Each step of the trajectory $(s_t, v_t, N(s_t))$ is encoded into the following pair of messages:

(a) An assistant message
$$\texttt{SELECTED\_STATE: } s_t$$

(b) A user message

$$\texttt{FEEDBACK\_SCORE for STATE } s_t: \quad v_t$$
$$\texttt{CHILD\_STATES for STATE } s_t: \quad s_{t,0} \; s_{t,1} \; ...$$

We then let the LLM generate the next assistant message and parse out the newly selected state.

```
You are a Tree-Search Assistant.  Your mission is to locate, within a fixed number of
iterations, the leaf node with the highest reward.  You'll follow a strict, turn-by-turn
protocol with the user.

### 1.  Initialization
- When the user's first message is 'START', immediately reply:  SELECTED_STATE: 0
- State 0 is the root.

### 2.  Turn Format
On each subsequent turn, the user will send information about the most recently selected state
N:
1.  FEEDBACK_SCORE for STATE: <score>
- This score is a stochastic estimate of the **average reward** of the leaf nodes reachable from
this state.
2.  CHILD_STATES for STATE N: <c1> <c2> ...  <ck>
- If there are no IDs after 'CHILD_STATES', then N is a leaf.

**Important:**
- The FEEDBACK_SCORE is a stochastic estimate of the average reward of the leaf nodes reachable
from this state.
- You only learn a state's score when you select it.
- You do not know the scores of its children or any other unselected states until you select
them.

### 3.  Your Selection Rule
- At each turn, pick exactly one unvisited state from any of the child-state lists you've
received so far (including lists from previous turns).
- Never select a leaf node that you have already visited.
- Reply with only:
SELECTED_STATE: X where X is the ID of the new state you're choosing.

### 4.  Objective
- Multiple leaf nodes have non-zero rewards; your goal is to find the one with the highest
reward.
- Use all received FEEDBACK_SCORES and CHILD_STATES data to guide your choices.
- Aim to identify the highest-reward leaf node in as few selections as possible.

### 5.  Strict Output Format
- Only output lines of the form SELECTED_STATE: X.
- Do not echo user input, add commentary, or output anything else.

Begin when the user says START.
```

Figure 13: System prompt used to drive the Tree-Search Assistant.

### C.4 Trace Format for Academic Paper Search Problem

In experiments on the Academic Paper Search problem, we use pretrained LLMs with the system prompts shown in Figures 14 and 15. At each iteration, the LLM selects the next paper.

For all methods, we evaluate the entire path as a single sequence and ask LLMs to assign one suitability score between 0.00 and 1.00 using the prompt in Figure 14. For the method without external search, we instead ask the LLM to select the single most suitable candidate paper for reaching the target paper, using the prompt in Figure 15.

```
You are an AI assistant specialized in finding a navigable academic paper from a given start
paper {start_title} to a target paper {target_title}.

1. Input:

    • PATH: The previously selected sequence of papers.
      Example:  PATH: "Paper A" -> "Paper B" -> "Paper D" -> ...

2. Task:

    • Evaluate the entire PATH as a whole.
    • Assign one suitability score from 0.00 to 1.00 indicating how appropriate the current PATH
      is for reaching {target_title}.

3. Output:

    • On the last line only, produce a single "score" and its value with two decimal places.
      Example:  0.76
    • No extra text or explanation.
```

Figure 14: System prompt used to score the path in the Academic Paper Search problem.

```
You are an AI assistant specialized in finding a navigable academic paper from a start paper to
a target paper.

1. Input:

    • PATH: The previously selected sequence of papers.
      Example:  PATH: "Paper A" -> "Paper B" -> "Paper D" -> ...
    • SCORE_LIST: A list of the scores for the corresponding papers in PATH.
      Example:  SCORE_LIST: 0.43 -> 0.22 -> 0.83 -> ...
    • CANDIDATE_LIST: A 1-based indexed list of candidate papers.
      Example:  CANDIDATE_LIST: 1.  "Paper A", 2.  "Paper B", 3.  "Paper C", ...

2. Task:

    • Balance exploration and exploitation when selecting the next paper from CANDIDATE_LIST.
    • Select the single most suitable paper for reaching the target paper.
    • Do not estimate any candidate as the target itself.

3. Output:

    • On the last line only, output exactly one integer:  the index of the chosen paper from
      CANDIDATE_LIST.
      Example:  2
    • No extra text or explanation.
```

Figure 15: System prompt used to select the candidate in the Academic Paper Search problem.

# D Theoretical Constructions of Transformers

In this section, we provide the details of our theoretical results missing in Section 4.

## D.1 Theoretical Transformer Model

This section defines a simplified version of the Transformer architecture used in this work. We make similar assumptions as in (Hahn, 2020; Weiss et al., 2021). We omit layer normalization and replace the fully connected layer with any arbitrary token-wise function (since MLPs are universal approximators). We also replace the conventional softmax attention mechanism with hard attention.

**Transformer Block** Let $X \in \mathbb{R}^{d \times n}$ be the input matrix representing a sequence of $n$ token embeddings, each of dimension $d$. A single-head Transformer *block* consists of a residual self-attention mechanism followed by a position-wise feed-forward function $f$:

First, the self-attention mechanism updates the input $X$:

$$\text{Attn}(X) := X + VX \ \text{hardmax}(X^\top K^\top QX) \quad \in \mathbb{R}^{d \times n},$$

Here, $Q, K, V \in \mathbb{R}^{d \times d}$ are the learnable query, key, and value weight matrices, respectively. The term $X^\top K^\top QX$ calculates the $n \times n$ attention weight matrix, where entry $(i, j)$ represents the attention from token $j$ to token $i$ and we define the hardmax operator as follows:

$$\text{hardmax}(z)_i := \frac{\mathbb{1}[z_i = \max_k z_k]}{\sum_j \mathbb{1}[z_j = \max_k z_k]}.$$

This operator assigns a uniform probability distribution over the index (or indices) corresponding to the maximum value(s) in $z$, and zero otherwise. Next, a feed-forward function $f$ is applied independently to each token's representation (each column vector):

$$\text{Block}(X) := \text{Attn}(X) + \left[ f\left(\text{Attn}(X)_{:,j}\right) \right]_{j=1}^n \quad \in \mathbb{R}^{d \times n},$$

where:

- $\text{Attn}(X)_{:,j}$ denotes the $j$-th column (token representation) after the attention step;

- $f : \mathbb{R}^d \to \mathbb{R}^d$ is any arbitrary piecewise continuous function. The output of $f$ for each token is added residually to the output of the attention layer.

**Transformer Network** A Transformer network is formed by stacking $\ell$ such blocks. Given an initial input sequence $X$ and fixed positional embeddings $p_1, p_2, \ldots, p_n \in \mathbb{R}^d$, we first incorporate positional information:

$$X^{(0)} = X + [p_1, p_2, \ldots, p_n].$$

Then, we iteratively apply the blocks:

$$X^{(k)} = \text{Block}\left(X^{(k-1)}\right), \quad \text{for } k = 1, \ldots, \ell.$$

A final linear layer (the "unembedding" matrix) $U \in \mathbb{R}^{v \times d}$ is applied to the final representation of the last token in the sequence, $X^{(\ell)}_{:,n}$, where $v$ is the vocabulary size:

$$\text{logits} = U X^{(\ell)}_{:,n} \in \mathbb{R}^v,$$

The predicted next token is the one corresponding to the highest logit value:

$$\text{next token} = \arg\max_i (\text{logits}_i),$$

i.e., we do greedy decoding. We break ties uniformly at random.

**Discussion on Model Assumptions**   The omission of layer normalization follows established practice in theoretical Transformer constructions including universal approximation proofs (Yun et al., 2020) and constructing programming languages that can be compiled into Transformers (Weiss et al., 2021).

Our hard attention assumption is mild in that it can be approximated arbitrarily closely by softmax attention through logit scaling. It is also a standard assumption used in previous Transformer complexity analyses (Pérez et al., 2019).

Our requirement for feed-forward networks to represent arbitrary piecewise-continuous functions aligns with classical universal approximation theorems. Specifically, ReLU network have been shown to be able to approximate any function in Sobolev spaces (Yarotsky, 2017) with tight bounds on the parameters needed to achieve arbitrary closeness.

Lastly, we note, these assumptions match those in the RASP programming model (Weiss et al., 2021), which similarly removes normalization layers and assumes full expressivity of feed-forward layers.

### D.2   Proof of Theorem 1

**Embedding Construction**   We embed each token into a vector whose coordinates ("registers") store interpretable features. Table 2 lists the registers, their meaning, and the initial values for each token type. The $\mathtt{id}_i$ register is a collection of registers, each corresponding to one state token $S_i$. The last three registers ($\mathtt{isVisited}$, $\mathtt{inhValue}$, and $\mathtt{pos}$) are initialized to zero and will be updated by the Transformer. Because, we require a register for each state, it follows that embedding dimension is size $\Theta(TB)$.

**Notation**   We write $e_{\mathtt{a}}$ for the standard basis vector with an 1 in the coordinate corresponding to register $\mathtt{a}$ and 0 elsewhere. For registers $\mathtt{a}$ and $\mathtt{b}$, define

$$M_{\mathtt{a}\to\mathtt{b}} = e_{\mathtt{b}}\, e_{\mathtt{a}}^{\top},$$

which is the matrix that copies the value from register $\mathtt{a}$ into register $\mathtt{b}$.

Table 2: Initial embeddings. Each row is a register (coordinate), and each column gives its initial value for the token types $V_\alpha$ (value), "$\mathtt{\#}$ $\mathtt{?}$ $\mathtt{\%}$" separators, and state tokens $S_k$.

| Register | $V_\alpha$ | $\mathtt{\#}$ | $\mathtt{?}$ | $\mathtt{\%}$ | $S_k$ |
|---|---|---|---|---|---|
| $\mathtt{value}$ | $\alpha$ | 0 | 0 | 0 | 0 |
| $\mathtt{\#?}$ | 0 | $+1$ | $-1$ | 0 | 0 |
| $\mathtt{isValue}$ | 1 | 0 | 0 | 0 | 0 |
| $\mathtt{is\#?}$ | 0 | 1 | 1 | 0 | 0 |
| $\mathtt{isState}$ | 0 | 0 | 0 | 1 | 0 |
| $\mathtt{id}_i$ | 0 | 0 | 0 | 0 | $\mathbb{1}[i=k]$ |
| $\mathtt{bias}$ | 1 | 1 | 1 | 1 | 1 |
| *Dynamically updated registers:* | | | | | |
| $\mathtt{isVisited}$ | 0 | 0 | 0 | 0 | 0 |
| $\mathtt{inhValue}$ | 0 | 0 | 0 | 0 | 0 |
| $\mathtt{pos}$ | 0 | 0 | 0 | 0 | 0 |

We also add a positional embedding by setting

$$X_t^{(0)} \leftarrow X_t + t\, e_{\mathtt{pos}},$$

so that the $\mathtt{pos}$ register of the token at position $t$ stores its index in the trace.

**Layer 1: Marking Visited States**   We want each state token to know whether it follows a $\mathtt{?}$ (i.e., was selected by the Transformer for selection) or if it was generated by the environment to indicate the children states.

Choose

$$Q = M_{\texttt{bias}\to 1}, \quad K = M_{\texttt{is\#?}\to 1}, \quad V = M_{\texttt{\#?}\to\texttt{isVisited}}.$$

Here `bias` queries uniformly, $K$ gives score 1 to separators (tokens with `is#?=1`), and $V$ copies the separator's `#?` (+1 for `#`, −1 for `?`) into the `isVisited` register of the querying token. Since a state immediately after `?` sees one more −1 than +1, its `isVisited` becomes negative; after `#` it sees equal counts and remains zero. Thus `isVisited` < 0 exactly for visited states.

We then apply a feed-forward that scales the `isValue` register of value tokens by their index in the trace:

$$f^{(1)}(y) = \left(y_{\texttt{isValue}}\, y_{\texttt{pos}}\right) e_{\texttt{isValue}},$$

which will be used in subsequent layers so that state tokens attend to only the closest preceding value token.

**Layer 2: Propagating Inherited Rewards**  Next we want each *frontier* state (states that are children of visited states but are themselves not yet visited) to inherit the rollout estimate value of its parent state. Parent values live in the most recent value token $V_\alpha$ before the state's position.

Set

$$Q = M_{\texttt{bias}\to 1}, \quad K = M_{\texttt{isValue}\to 1}, \quad V = M_{\texttt{value}\to\texttt{inhValue}}.$$

Each state token attends (via hard attention) to the immediately preceding value token (due to the feedforward in the previous layer), and the $V$ matrix writes that value $\alpha$ into its `inhValue` register.

We then use a piecewise feed-forward on the $\texttt{id}_i$ registers:

$$f^{(2)}(y)_{\texttt{id}_i} = \begin{cases} y_{\texttt{inhValue}}, & \text{if,} y_{\texttt{isVisited}} = 0 \\ -1, & \text{otherwise,} \end{cases}$$

with all other coordinates set to zero. Non-state tokens and visited state tokens get a negative score, and each frontier state $S_i$ has its inherited reward stored in $\texttt{id}_i$.

**Layer 3: Selecting the Maximum**  Finally, we collect all frontier scores and choose the largest:

$$Q = M_{\texttt{bias}\to 1}, \quad K = M_{\texttt{isState}\to 1}, \quad V = \sum_i M_{\texttt{id}_i\to\texttt{id}_i}.$$

At the embedding corresponding to the last `?` in the trace, the query attends to all preceding state tokens (`isState=1`) and $V$ sums each state $S_i$'s $\texttt{id}_i$ value into the querying token's $\texttt{id}_i$. We set $f^{(3)}(y) = 0$ as we do not need the feedforward in this layer. Now assuming tokens are indexed so that the embedding coordinate $\texttt{id}_i$ corresponds exactly to the vocabulary index of state token $S_i$, we choose the unembedding matrix $U$ to be zero everywhere except on the submatrix mapping the $\texttt{id}_i$ coordinates to their corresponding token logits, where it is the identity. In this setup, the $\texttt{id}_i$ registers directly become the logits for each state token, so greedy decoding picks the state with the highest inherited reward.

**Remark (Uniform-Leaf Sampling)**  If instead all $V_\alpha$ embeddings are set to the same constant vector, then every frontier state receives an identical score in Layer 2, yielding uniform sampling over leaves by the same construction.

## D.3  Proof of Theorem 2

For this theorem, we analyze the conventional MCTS (Sutton & Barto, 2018), which is a widely-used form, where the selected state $s^*$ is determined by taking the $\arg\min$ over $N(s)$, rather than $N^*(s)$ as in Algorithm 6. In our empirical experiments, we used $N^*(s)$ due to the shallow nature of the search trees. However, for theoretical clarity and consistency with classical MCTS, the Transformer construction presented here uses $N(s)$.

**Embedding Construction**  Tokens are again embedded into interpretable "registers." Table 3 defines the initial static register values; all dynamic registers are initialized to zero. Registers of type $\mathtt{X}_i$ denote a set $\{\mathtt{X}_i\}_{i=1}^{TB+1}$, with one register per state token. The $\mathtt{oid}_i$ registers include two additional coordinates used specifically for the $\mathtt{>}$ and $\mathtt{\%}$ tokens.

As in the previous construction, we apply a positional embedding via $X_t^{(0)} \leftarrow X_t^{(0)} + t\,e_{\mathtt{pos}}$, so that each token's $\mathtt{pos}$ register holds its position in the trace.

Table 3: Initial embeddings for the MCTS-Transformer: static registers encode each token's fixed features, while dynamic registers are initialized to zero and updated by the network conditioned on the trace.

| Register | $V_\alpha$ | # | ? | > | [bos] | % | $S_k$ |
|---|---|---|---|---|---|---|---|
| value | $\alpha$ | 0 | 0 | 0 | 0 | 0 | 0 |
| #? | 0 | +1 | −1 | 0 | 0 | 0 | 0 |
| isValue | 1 | 0 | 0 | 0 | 0 | 0 | 0 |
| is#? | 0 | 1 | 1 | 0 | 0 | 0 | 0 |
| is> | 0 | 0 | 0 | 1 | 0 | 0 | 0 |
| is? | 0 | 0 | 1 | 0 | 0 | 0 | 0 |
| isBos | 0 | 0 | 0 | 0 | 1 | 0 | 0 |
| is#Bos | 0 | 1 | 0 | 0 | 1 | 0 | 0 |
| isState | 0 | 0 | 0 | 0 | 0 | 0 | 1 |
| $\mathtt{id}_i$ | 0 | 0 | 0 | 0 | 0 | 0 | $\mathbb{1}[i=k]$ |
| bias | 1 | 1 | 1 | 1 | 1 | 1 | 1 |
| *Dynamically updated registers:* | | | | | | | |
| is?·pos | 0 | 0 | 0 | 0 | 0 | 0 | 0 |
| isValue·pos | 0 | 0 | 0 | 0 | 0 | 0 | 0 |
| isState·pos | 0 | 0 | 0 | 0 | 0 | 0 | 0 |
| closest?pos | 0 | 0 | 0 | 0 | 0 | 0 | 0 |
| parentpos | 0 | 0 | 0 | 0 | 0 | 0 | 0 |
| isVisited | 0 | 0 | 0 | 0 | 0 | 0 | 0 |
| wasVisited | 0 | 0 | 0 | 0 | 0 | 0 | 0 |
| $\mathtt{vid}_i$ | 0 | 0 | 0 | 0 | 0 | 0 | 0 |
| $\mathtt{cid}_i$ | 0 | 0 | 0 | 0 | 0 | 0 | 0 |
| $\mathtt{pid}_i$ | 0 | 0 | 0 | 0 | 0 | 0 | 0 |
| $\mathtt{psid}_i$ | 0 | 0 | 0 | 0 | 0 | 0 | 0 |
| $\mathtt{nsid}_i$ | 0 | 0 | 0 | 0 | 0 | 0 | 0 |
| $\mathtt{oid}_i$ | 0 | 0 | 0 | 0 | 0 | 0 | 0 |
| iter | 0 | 0 | 0 | 0 | 0 | 0 | 0 |
| pos | 0 | 0 | 0 | 0 | 0 | 0 | 0 |

For any coordinates where we do not explicitly define how it is updated by a feedforward function, the output is defined to be zero.

**Layer 1: Iteration Counter and Positional Precomputation**  This layer precomputes positional dependent registers. The attention mechanism is defined by the following:

$$Q = M_{\mathtt{bias}\rightarrow 1},$$
$$K = M_{\mathtt{is\#bos}\rightarrow 1},$$
$$V = M_{\mathtt{isBos}\rightarrow \mathtt{iter}}.$$

This attention is used to compute the number of iterations that have passed which will be stored in the `iter` register. We apply the feed-forward function $f^{(1)}$:

$$f^{(1)}(y)_{\texttt{is?P}} = y_{\texttt{is?}} y_{\texttt{pos}},$$
$$f^{(1)}(y)_{\texttt{isState·pos}} = y_{\texttt{isState}} y_{\texttt{pos}},$$
$$f^{(1)}(y)_{\texttt{isValue·pos}} = y_{\texttt{isValue}} y_{\texttt{pos}},$$
$$f^{(1)}(y)_{\texttt{iter}} = (1/y_{\texttt{iter}}) - y_{\texttt{iter}}, \quad \text{if } y_{\texttt{iter}} > 0 \text{ else } 0,$$

which processes the `iter` register to hold the correct value and preprocess several other positionally dependent registers for later computation.

**Layer 2: Compute `closest?pos`**  This layer will do computation so that when `isState` is 1, `closest?pos` will store the position of the closest preceding `?` token in the trace. Otherwise, it is set to 0. The attention matrices are given by the following:

$$Q = M_{\texttt{bias}\to 1},$$
$$K = M_{\texttt{is?·pos}\to 1},$$
$$V = M_{\texttt{pos}\to\texttt{closest?pos}}.$$

Through the hard attention, each token attends to the preceding token with the largest `is?·pos` value (i.e., the value of the closest preceding `?` token). Its `pos` value is copied to the querying token's `closest?pos` register. The subsequent feed-forward function $f^{(2)}$ is:

$$f^{(2)}(y)_{\texttt{closest?pos}} = y_{\texttt{closest?pos}}\, y_{\texttt{isState}} - y_{\texttt{closest?pos}},$$

which zeros out `closest?pos` if the current token does not correspond to a state.

**Layer 3: Compute Per-Iteration Reward and Count Statistics**  This layer computes the reward and tree policy visitation count statistics for each iteration, which will be stored in the respective $V_\alpha$ token at the end of its iteration. The attention is:

$$Q = M_{\texttt{bias}\to 1},$$
$$K = M_{\texttt{closest?pos}\to 1},$$
$$V = \sum_i (M_{\texttt{id}_i\to\texttt{vid}_i} + M_{\texttt{id}_i\to\texttt{cid}_i}).$$

For a $V_\alpha$ token, this attention sums statistics from state tokens $s_k$ visited by the tree policy within the current iteration, identified as the tokens with the largest value in `closest?P`. The $\texttt{id}_i$ values from these $s_k$ tokens are copied and summed into the $V_\alpha$ token's $\texttt{vid}_i$ and $\texttt{cid}_i$ registers. The feed-forward function $f^{(3)}$, applied at $V_\alpha$ tokens, then sets:

$$f^{(3)}(y)_{\texttt{vid}_i} = y_{\texttt{value}}\, \mathbb{1}[y_{\texttt{vid}_i} > 0] - y_{\texttt{vid}_i},$$
$$f^{(3)}(y)_{\texttt{cid}_i} = \mathbb{1}[y_{\texttt{cid}_i} > 0] - y_{\texttt{vid}_i},$$

storing 1 in $\texttt{cid}_i$ when state $S_i$ is visited in this iteration (used to compute visitation counts).

**Layer 4: Aggregate Statistics (Backpropagation)**  This layer aggregates reward and count statistics from all iterations. These accumulated statistics will be stored in the final $V_\alpha$ token of the trace. We set:

$$Q = M_{\texttt{bias}\to 1},$$
$$K = M_{\texttt{isValue}\to 1},$$
$$V = \sum_i (M_{\texttt{vid}_i\to\texttt{vid}_i} + M_{\texttt{cid}_i\to\texttt{cid}_i}).$$

The token intended for final aggregation attends to all previous $V_\alpha$ tokens (where $\texttt{isValue}= 1$). It sums up the $\texttt{vid}_i$ (values) and $\texttt{cid}_i$ (counts) from these iteration-specific $V_\alpha$ tokens into its own registers. We then use the feedforward layer to unnormalize by the number of iterations (since the attention will average them):

$$f^{(5)}(y)_{\texttt{cid}_i} = y_{\texttt{cid}_i}\, y_{\texttt{iter}} - y_{\texttt{cid}_i}$$
$$f^{(5)}(y)_{\texttt{vid}_i} = y_{\texttt{vid}_i}\, y_{\texttt{iter}} - y_{\texttt{vid}_i}.$$

**Layer 5: Identify State Tokens Selected by Transformer**   This layer identifies which state tokens in the trace corresponds to those generated by the Transformer during the tree policy trace. We use attention:

$$Q = M_{\texttt{bias}\rightarrow 1},$$
$$K = M_{\texttt{is?\#}\rightarrow 1},$$
$$V = M_{\texttt{?\#}\rightarrow\texttt{isSelected}}.$$

Each state token sums the `?#` values from preceding relevant tokens. For states immediately following a `#`, this sum in `isSelected` will be 0. It will be nonzero otherwise. The feed-forward $f^{(5)}$ uses this fact to compute the register:

$$f^{(5)}(y)_{\texttt{isSelected}} = y_{\texttt{isState}} \cdot \mathbb{1}\left[y_{\texttt{isSelected}} \neq 0\right] - y_{\texttt{isSelected}}.$$

This sets `isSelected` to 1 for state tokens selected by the Transformer and 0 for the state token generated by the environment to indicate the child nodes.

**Layer 6: Identify Parent Position for Neighbors**   This layer serves as preprocessing to identify the parent node of each neighbor token (i.e., those state tokens immediately succeeding the `#` token), by first storing the parent's position in the `parentpos` register of the parent token itself. No attention is needed, so we set all the attention weights to 0. The feed-forward $f^{(6)}$ is:

$$f^{(6)}(y)_{\texttt{parentP}} = y_{\texttt{isState}} \cdot y_{\texttt{isSelected}} \cdot y_{\texttt{pos}}.$$

This operation stores the current token's position into its own `parentP` register if it is a state token and corresponds to a state visited by the tree policy in some iteration.

**Layer 7: Propagate Parent `id` to Its Children States**   Children state tokens use the information about the parent's position to retrieve the parent's `id`. The attention is:

$$Q = M_{\texttt{bias}\rightarrow 1},$$
$$K = M_{\texttt{parentpos}\rightarrow 1},$$
$$V = \sum_i M_{\texttt{id}_i\rightarrow\texttt{pid}_i}.$$

Each neighbor state token attends to the immediately preceding non-neighbor state (which will be it's parent state). It then copies the parent token's $\texttt{id}_i$ into its own $\texttt{pid}_i$ registers. No feed-forward $f^{(7)}$ is needed, so we set it to 0.

**Layer 8: Store Previously Selected State `id` in Embedding of `>` Token**   The `>` token stores the `id` of the state that was just selected by the tree policy. We set:

$$Q = M_{\texttt{bias}\rightarrow 1},$$
$$K = M_{\texttt{isState}\cdot\texttt{pos}\rightarrow 1},$$
$$V = \sum_i M_{\texttt{id}_i\rightarrow\texttt{psid}_i}.$$

The `>` token attends to the immediately preceding state token and copies that state's $\texttt{id}_i$ into its own $\texttt{psid}_i$.

**Layer 9: Collect Children States of Previously Selected State in Embedding of > Token**   The
> token collects the `id` of all children of the previously selected state. The attention is:

$$Q = \sum_i M_{\mathtt{psid}_i \to i},$$
$$K = \sum_i M_{\mathtt{pid}_i \to i},$$
$$V = \sum_i M_{\mathtt{id}_i \to \mathtt{nsid}_i}.$$

The > token uses its $\mathtt{psid}_i$ registers as the query. State tokens use their $\mathtt{pid}_i$ registers as keys. If key and
query match, the neighbor's $\mathtt{id}_i$ is copied/summed into the > token's $\mathtt{nsid}_i$ register. No feed-forward $f^{(9)}$ is
needed, so we set it to 0.

**Layer 10: Final UCT Computation for > Token**   At the > token, the next state to select is calculated
using UCT scores. First, it gathers aggregated statistics using the following attention weights:

$$Q = M_{\mathtt{bias} \to 1},$$
$$K = M_{\mathtt{isValue} \cdot \mathtt{pos} \to 1},$$
$$V = \sum_i (M_{\mathtt{cid}_i \to \mathtt{cid}_i} + M_{\mathtt{vid}_i \to \mathtt{vid}_i}).$$

The > token attends to the token holding globally aggregated statistics (computed in layer 4) into its own
registers. The feed-forward function $f^{(10)}$ then does the UCT computation:

$$f^{(10)}(y)_{\mathtt{oid}_i} = \begin{cases} y_{\mathtt{vid}_i} + c\sqrt{\log(2\sum_k y_{\mathtt{cid}_k} y_{\mathtt{psid}_k})/y_{\mathtt{cid}_i}} & \text{if } y_{\mathtt{is>}} = 1,\ y_{\mathtt{nsid}_i} = 1 \text{ and } y_{\mathtt{cid}_i} \neq 0, \\ \infty, & \text{if } y_{\mathtt{is>}} = 1,\ y_{\mathtt{nsid}_i} = 1 \text{ and } y_{\mathtt{cid}_i} = 0, \\ 0, & \text{otherwise.} \end{cases}$$

**Layer 11: Generate Start State $S_0$ After ?**   This layer ensures the Transformer generates $S_0$ (the root
node) when it sees the initial ? token. The attention matrices are

$$Q = 0,$$
$$K = 0,$$
$$V = M_{\mathtt{is?} \to \mathtt{oid}_0}.$$

We set $f^{(11)}$ to the 0 function in this layer.

**Layer 12: Generate > or % After State Selection**   This layer determines whether to generate > or %
after a state token has been chosen by the tree policy. Specifically > is generated if the tree policy has not
reached a new undiscovered node. The % token is generated otherwise. For this layer, we set the attention
matrices to the following:

$$Q = \sum_i M_{\mathtt{id}_i \to i},$$
$$K = \sum_i M_{\mathtt{id}_i \to i},$$
$$V = M_{\mathtt{isSelected} \to \mathtt{wasSelected}}.$$

Intuitively for the current state token in the tree policy path, it will attend to all previous state tokens
that represent the same states. It aggregates the `isSelected` values into the `wasSelected` register. Thus
`wasSelected` $> 0$ if and only if the state was previously visited in another iteration. Thus we let the feed
forward layer be

$$f^{(12)}(y)_{\mathtt{oid}_>} = y_{\mathtt{isState}} \mathbb{1}[y_{\mathtt{wasVisited}} > 0],$$
$$f^{(12)}(y)_{\mathtt{oid}_\%} = y_{\mathtt{isState}} \mathbb{1}[y_{\mathtt{wasVisited}} = 0].$$

**Final Unembedding** Assuming without loss of generality that $\mathtt{oid}_i$ corresponds to the $i$'th state token, it again suffices to let $U$ be the matrix that is zero everywhere except on the submatrix mapping the $\mathtt{oid}_i$ to the corresponding token. The token with the highest logit value is chosen by greedy decoding.

**Note on on Different Tree Policies** For path sampling with a greedy policy and uniform path sampling, Layer 10 is modified. For path sampling with a greedy policy, the feed-forward $f^{(10)}$ at the `>` token computes:

$$f^{(10)}(y)_{\mathtt{oid}_i} = y_{\mathtt{vid}_i}\, y_{\mathtt{is>}}\, y_{\mathtt{nsid}_i}.$$

For uniform path sampling, $f^{(10)}$ would set:

$$f^{(10)}(y)_{\mathtt{oid}_i} = y_{\mathtt{is>}}\, y_{\mathtt{nsid}_i}.$$

# E  Details of Empirical Analysis on Transformers' Search Capabilities

In this work, we use NumPy (Harris et al., 2020), SciPy (Virtanen et al., 2020), PyTorch (Paszke et al., 2019), Scikit-learn (Pedregosa et al., 2011), and other scientific packages in Python. Moreover, as shown in the main article, we employ a variety of LLM APIs and checkpoints such as OpenAI GPT, Google Gemini, and Alibaba Qwen.

To run our experiments, we utilize commercial Intel and AMD CPUs such as AMD EPYC 9374F and Intel Xeon Platinum 8352Y, and NVIDIA GPUs such as NVIDIA L40S.

## E.1  Experimental Details of Pretrained LLMs on Multi-Reward Tree Search Problem

To test the performance of pretrained LLM's on the Multi-Reward Tree Search problem, we use the prompting method as in Section C.3. We use a tree-depth of 6 and branching factor of 2. From among all leaf nodes, we uniformly sample 8 distinct leafs to serve as "goal" nodes. Each selected goal leaf is then randomly assigned a unique reward from the set $\{0.1, 0.2, 0.3, 0.4, 0.5, 0.6, 0.7, 1.0\}$. In total, we generate 200 independent trees according to this procedure, resulting in 200 distinct multi-reward tree search instances.

**Model Evaluation**  We evaluate the following models using the prompting strategy described above: GPT-4.1-mini and GPT-4.1 (OpenAI, 2025a); Gemini 2.5-Flash and Gemini 2.5-Flash (Thinking) (Gemini Team, Google, 2025); and Qwen3-8B and Qwen3-8B (Thinking) (Qwen Team, 2025). Each model is tested for 50 steps per tree instance.

**Model Configuration**  The "Thinking" variants refer to configurations in which explicit reasoning modes are activated. For Gemini 2.5-Flash (Thinking), we set the `reasoning_effort` parameter to `medium`. For Qwen3-8B, which by default engages in "Thinking" (i.e., explicit reasoning chains), we create a non-thinking variant by appending `/no_think` to the beginning of the system prompt to suppress intermediate reasoning steps. Lastly, we use greedy decoding for all variants.

### E.2 Experimental Details of Fine-Tuning for Enhancing Search Capabilities of LLMs

We evaluate the search abilities of pretrained and fine-tuned LLMs on the Academic Paper Search problem. Given a start paper, the goal of this problem is to find a target paper where the child states of a particular paper are defined by the papers of the co-authors of that particular paper. We randomly choose pairs of start and target papers as shown in Table 4. We employ OpenAlex (Priem et al., 2022) for designing this experiment. The prompts used to generate the search trajectories in these experiments are shown in Section C.4.

We fine-tune the Qwen3-8B model (Qwen Team, 2025) using Low-Rank Adaptation (LoRA) (Hu et al., 2022), a parameter-efficient approach that updates only a small set of parameters. The maximum sequence length is set to 10,240, and training is run for one epoch. We use the paged AdamW (32-bit) optimizer (Loshchilov & Hutter, 2018) for memory efficiency, with a learning rate of $2 \times 10^{-4}$, a cosine learning-rate schedule, 3% warmup, and weight decay 0.001. LoRA is configured with rank 16, scaling factor 32, and dropout 0.05, and is applied to the main attention projections and MLP projections.

Table 4: Start–target paper pairs in the Academic Paper Search problem.

| Start Paper | Target Paper | Start Paper | Target Paper |
|---|---|---|---|
| W2585547904 | W3196632732 | W2390034408 | W2357505547 |
| W2272018300 | W574723991 | W3196681629 | W2780169483 |
| W4252577446 | W3213173784 | W133467238 | W3138433919 |
| W3023894403 | W3186777337 | W2887992654 | W3035676168 |
| W2610360324 | W2254883584 | W28024976 | W4230887124 |
| W375670644 | W2645342909 | W4403312580 | W2360871073 |
| W2290528378 | W2184245987 | W2075179071 | W2413728222 |
| W1497510206 | W193222316 | W1928336576 | W4294761907 |
| W2378250669 | W2167634255 | W70597564 | W2019636961 |
| W2413184738 | W2994021882 | W2111031504 | W1538861733 |
| W3125578107 | W4205416125 | W1999318920 | W1648404904 |
| W957051818 | W1496033586 | W1043586152 | W1003374900 |
| W4225958650 | W4322489421 | W1983902283 | W2800480410 |
| W657940841 | W4387063953 | W2124237366 | W1986139416 |
| W2564522209 | W2246496635 | W2312519939 | W2263184888 |
| W3038306494 | W3041988068 | W2367761479 | W2935041779 |
| W2272635734 | W3203890190 | W2363737347 | W2368854302 |
| W204856840 | W2136708823 | W2330377032 | W2045968954 |
| W4312079109 | W3082198414 | W2290947939 | W3182725060 |
| W2053134222 | W4404914451 | W639808865 | W658822377 |
| W4322827068 | W2106442077 | W2316765899 | W4380875192 |
| W4242477155 | W4205383455 | W2415588074 | W3013098164 |
| W1525894204 | W1538920755 | W3173752466 | W2952700754 |
| W270839233 | W385335894 | W2008748940 | W2140973411 |
| W2090948737 | W2080692806 | W4392218955 | W4391238054 |
| W2009460780 | W2505367746 | W46357476 | W2314777899 |
| W2491706897 | W4232504235 | W1461752455 | W1540011986 |
| W2120932789 | W2962025900 | W2372889019 | W2355250576 |
| W575771764 | W427966303 | W2372242901 | W2998013669 |
| W2169084069 | W4291372033 | W4410927769 | W4403703781 |
| W2394974804 | W4256391018 | W3209857023 | W2001189511 |
| W2369940005 | W2356021002 | W2393662227 | W2358750724 |
| W3153204805 | W3153990973 | W2134325687 | W2337799504 |

# F   Additional Empirical Results

In this section, we show additional empirical results on the multi-reward tree search and multi-reward navigation problems.

## F.1   Additional Empirical Results on Multi-Reward Tree Search

In addition to the experiments shown in Figure 3, we present additional results on multi-reward tree search problems with more diverse settings.

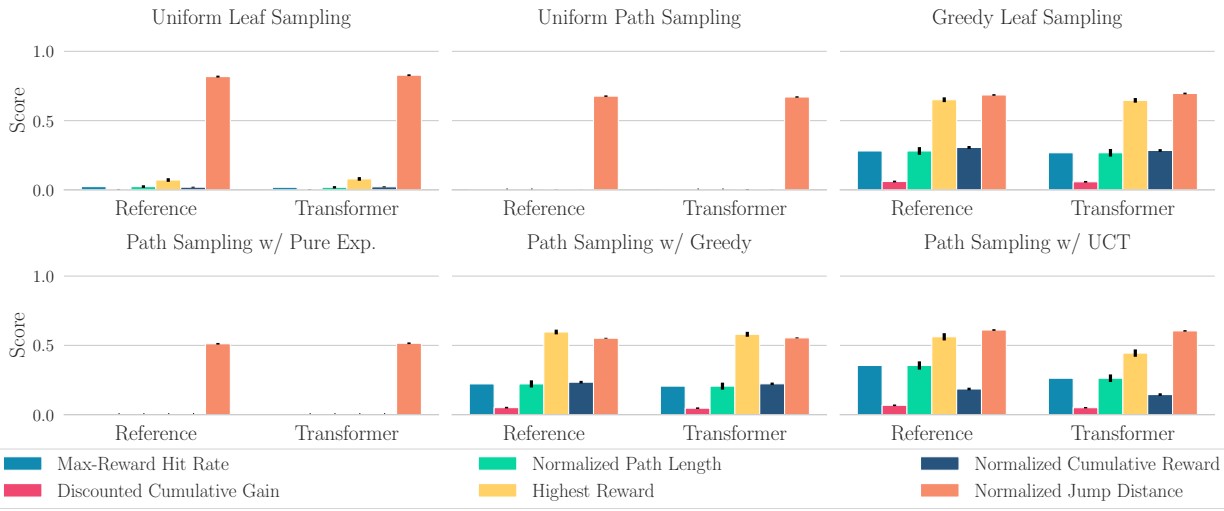

Figure 16: Behavior cloning results on the multi-reward tree search problem, where each binary tree of depth 8 has 8 different goals and a search step budget is 50.

Figure 16 shows the experimental results that test deeper multi-reward trees, which are depth 8. These results follow our observation that Transformers successfully mimic the behaviors of reference search algorithms. Since this setting is harder than the setting of Figure 3, the search performance of each algorithm is relatively poor.

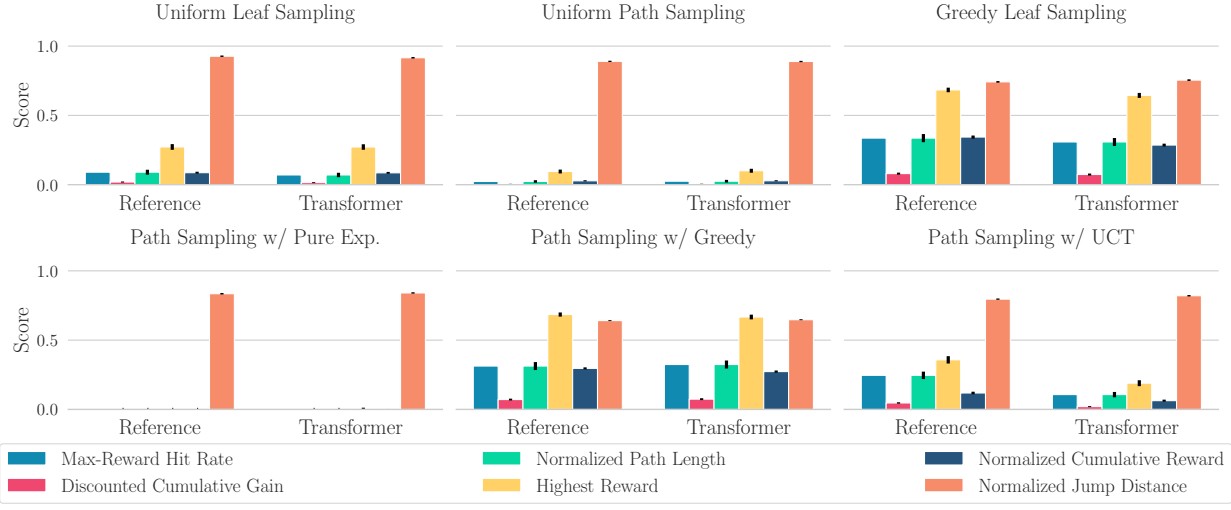

Figure 17: Behavior cloning results on the multi-reward tree search problem, where each quaternary tree of depth 4 has 8 different goals and a search step budget is 50.

Similarly, we conduct experiments on wider multi-reward trees in Figure 17. These trees have 4 child states for each parent state. Transformers successfully show similar performance of the respective search strategies, following our expectation.

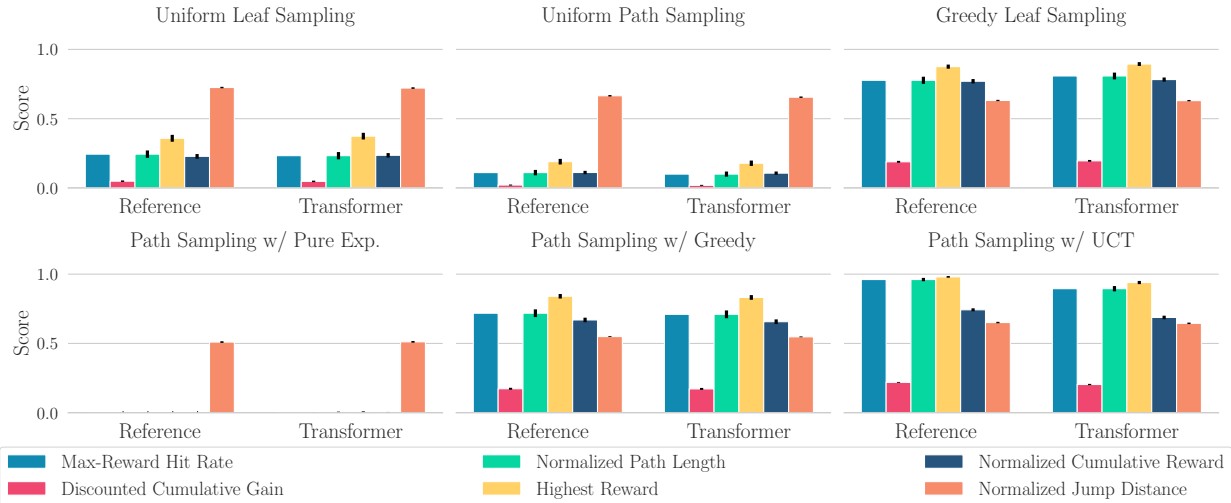

Figure 18: Behavior cloning results on the multi-reward tree search problem, where each binary tree of depth 6 has 4 different goals and a search step budget is 50.

We test a problem with a fewer number of goal sates. As shown in Figure 18, these results follow the findings observed in the previous experiments.

## F.2 Additional Empirical Results on Generalization Analysis

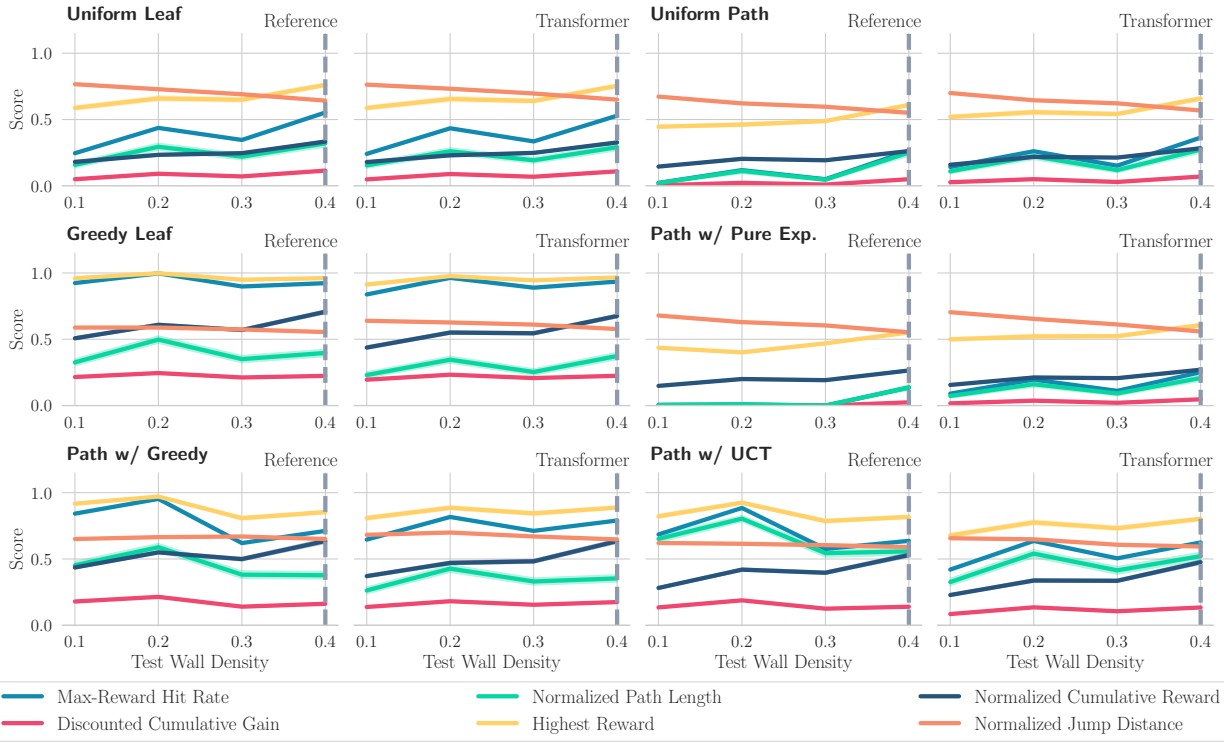

Figure 19: Generalization analysis over test wall densities on the multi-reward navigation problem of size 4 × 4 with a step budget of 50. The wall density smaller than 0.4 is unseen in a training phase. A gray dashed line indicates the setting used in training.

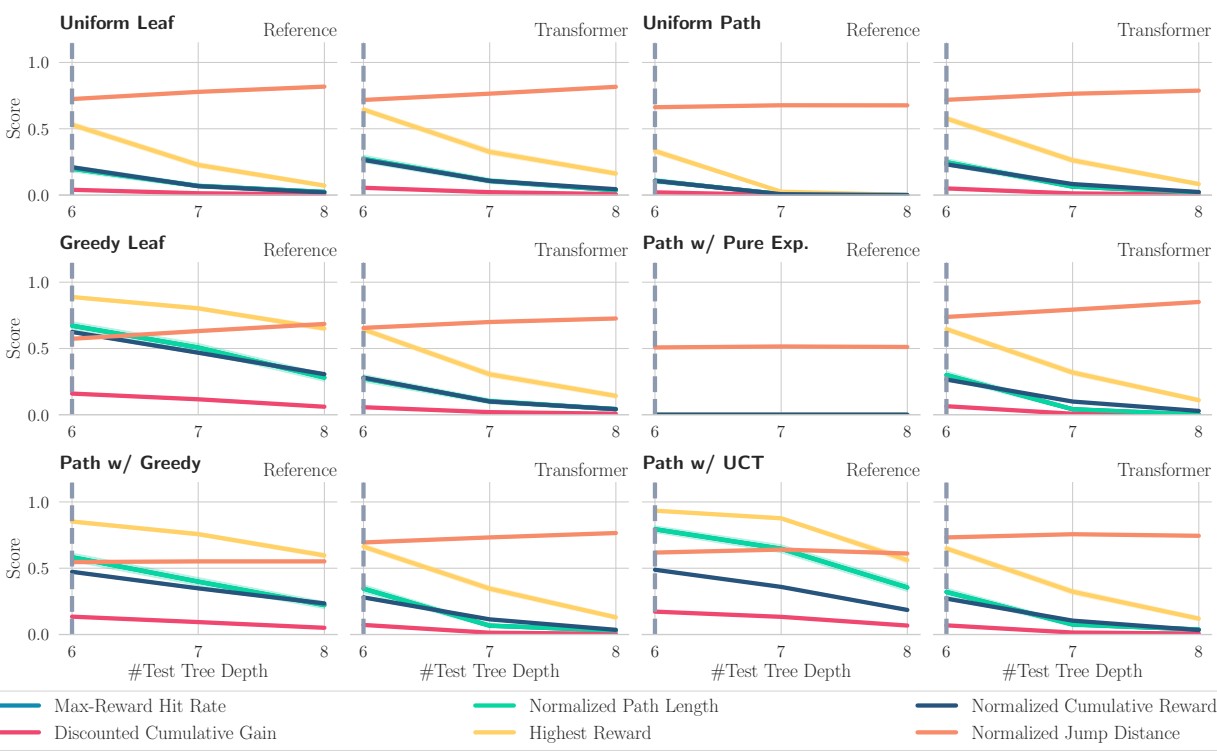

Figure 20: Generalization analysis over test tree depths on the binary tree search problem with 8 different goal states and a step budget of 50. The test tree depth larger than 6 is unseen in training. A gray dashed line indicates the setting used in training.

Figures 19 and 20 demonstrate generalization analysis over test wall densities, the numbers of test goal states, and test tree depths. In Figure 19, test cases with lower wall densities are more challenging than those with higher densities, as they tend to produce deeper and wider trees. Notably, according to Figure 20, generalizing to unseen tree depths is more challenging than other generalization settings. This can be seen as a challenge analogous to length generalization, similar to what is discussed in the previous work (Lee et al., 2024; 2025).

## G    Discussion on Safeguards

To promote reliability and reproducibility in our experiments, we aim to establish a set of safeguards focused on improving traceability and robustness. Our approach includes, in part, logging key inputs, outputs, and intermediate states, as well as verifying that the model behaves consistently under fixed conditions. While these components have been partially implemented in our current work, we consider them part of a broader, ongoing effort toward responsible model development. So far, we have not observed any misbehavior in our experiments.

## H    Declaration of LLM Usage

In addition to being the primary focus of our study, LLMs were used throughout our research process. We leveraged them to generate and refine system prompts for all experiments involving pretrained LLM models, assisted in writing the code used to run these experiments, and revised the manuscript itself. Specifically, we employed LLM platforms developed by OpenAI, Google Gemini, and Ollama.

