# OpenReview forum: "Transformers in the Dark: Navigating Unknown Search Spaces via Bandit Feedback"
_TMLR — Accepted by TMLR_

### Review · Reviewer_SX3A · 2025-12-12

**Summary Of Contributions:**

The paper investigates whether Transformers can internalize and execute structured search algorithms without reliance on external search modules. The authors propose a framework called "Unknown Tree Search with Bandit Feedback" to isolate the model's search policy from the environment dynamics.

The contributions are threefold:

- Theoretical Analysis: The authors provide constructive proofs demonstrating that Transformers of constant depth are theoretically expressive enough to implement various leaf-based and path-based search strategies.

- Empirical Verification (Behavior Cloning): Through experiments on synthetic tree and navigation tasks, the paper shows that Transformers trained from scratch can clone the behavior of reference algorithms, replicating both performance outcomes and behavioral patterns (e.g., jump distances).

- Targeted Fine-Tuning: The authors demonstrate that fine-tuning a pre-trained LLM (Qwen3-8B) on search trajectories significantly improves its performance on a realistic "Academic Paper Search" task compared to standard prompting baselines.


**Strengths**

- Clear Motivation and Accessible Narrative: The paper is well-written and easy to follow. The motivation is compelling: moving beyond simply "using" LLMs for reasoning to understanding if they can internalize the search process itself. The progression from theoretical proofs to synthetic experiments and finally to a realistic downstream task creates a logical and cohesive narrative structure.

- Effective Use of Synthetic Environments to Isolate Search Logic: A key strength is the introduction of the "Unknown Tree Search with Bandit Feedback" framework utilizing controlled synthetic environments. By externalizing state expansion and using abstract states, the authors effectively decouple the "Search Policy" (decided by the Agent) from the environment dynamics. This design is crucial for isolating the model's pure algorithmic ability to balance exploration and exploitation, ensuring that the evaluation is not confounded by the model's vast pre-trained background knowledge or semantic priors.

- Comprehensive Experimental Design on Synthetic Data: Chapter 5 provides a thorough empirical analysis. The authors go beyond simple performance metrics and include behavioral metrics such as KL divergence and normalized jump distance. This multi-faceted analysis convincingly demonstrates that the model is not just achieving similar results but is actually mimicking the process of the reference algorithms. Furthermore, the inclusion of generalization tests (varying wall densities, step budgets, etc.) adds robustness to the claims of behavior cloning.


**Weaknesses**

- Concerns Regarding Synthetic Data Scale and Generalization: My primary concern lies in the complexity of the synthetic environments. The experiments primarily rely on small-scale settings, such as binary trees of depth 6 or 4x4 grids. While the paper claims generalization capabilities, the results in Appendix Figure 20 paint a different picture. There is a significant performance drop when the tree depth increases from 6 (seen) to 8 (unseen). This sharp degradation questions the core claim that the Transformer has truly "internalized" the recursive search algorithm. It suggests the model might be relying on shallow pattern matching or local heuristics that fail to scale, rather than learning a generalizable search procedure.

- Lack of Detail and Motivation in Targeted Fine-Tuning (Chapter 6): The transition to the "Academic Paper Search" task in Chapter 6 feels somewhat disconnected from the rigorous synthetic analysis. It lacks crucial experimental details regarding the fine-tuning process. For instance, the size of the training dataset, the number of epochs are not clearly detailed in the main text. The motivation for this specific task setup needs strengthening. It is unclear why this specific domain was chosen as the primary validator for "real-world" performance over more standard reasoning benchmarks.

- Reliability of LLM-Generated Bandit Feedback: The real-world scenarios rely on the assumption that "we can prompt the LLM to directly estimate state values... These realizations serve as noisy bandit feedback signals". This introduces a critical unverified variable: How accurate are these LLM-generated values? If the "bandit feedback" is highly inaccurate or inconsistent, the search agent is effectively navigating with a broken compass. The paper focuses on the search policy but takes the reward signal for granted. A high search capability on top of unreliable value estimation does not yield a good solver. The paper should evaluate the correlation between the LLM's estimated values and the ground truth (or task success) to validate this setup

**Audience:**

Yes

**Audience Explanation:**

This is a highly relevant direction as it attempts to rigorously analyze the planning and reasoning limits of LLMs by isolating their search policy from their world knowledge using the "Unknown Tree Search with Bandit Feedback" framework. The paper provides a comprehensive analysis by combining theoretical proofs of the Transformer's expressivity to represent these algorithms with empirical evidence that they can be trained to clone complex search behaviors and generalize to unseen conditions.

**Claims And Evidence:**

Yes

**Claims Explanation:**

the claims made in the submission are supported by accurate, convincing, and clear evidence, particularly regarding the feasibility of Transformers internalizing search algorithms. The experimental design in Chapter 5 is careful, and the results provide robust support for the authors' central hypothesis.

- Isolation of Variables: By using synthetic environments rather than natural language tasks, the authors effectively eliminate confounding factors such as the model's pre-training knowledge or semantic hallucinations. This ensures that any observed performance is due to the learned search policy .

- Comprehensive Metrics: The authors go beyond simple success rates. They employ process-oriented metrics such as Normalized Jump Distance and KL Divergence. This design choice is crucial because it verifies that the model is not just achieving the same result by chance, but is actually replicating the behavioral patterns of the reference algorithms.

- Successful Behavior Cloning: The experiments demonstrate that Transformers trained from scratch can match the performance of specific reference algorithms with high precision. The low KL divergence for leaf-based strategies further confirms that the models can approximate the target probability distributions accurately.

- Evidence of Internalization: The analysis of Jump Distance provides convincing evidence that the models have internalized the structural traversal logic of the algorithms, distinct from random exploration.

- Clear Evidence of Generalization **(with Limitations)** The authors provide clear evidence that the learned behaviors are not merely memorized but generalized:

**Requested Changes:**

- Contextualize Generalization Limits (Critical): While the paper claims the models can generalize, the results in Appendix Figure 20  show a clear performance degradation when the tree depth increases from 6 to 8. This suggests that the "internalized search" might be limited to a finite horizon or specific pattern matching rather than a fully recursive algorithm.

  Request: Please explicitly discuss this degradation in the main text (Section 5 or Limitations). Analyze why the performance drops—is it due to context length limitations, attention dispersion, or a breakdown of the learned heuristic? The claims regarding generalization should be calibrated to reflect these empirical bounds (e.g., "generalizes within a local range" rather than broad generalization).

- Provide Missing Details for Targeted Fine-Tuning (Critical) Section 6 demonstrates that fine-tuning Qwen3-8B improves search capabilities, but lacks critical reproducibility details.

  Request: Please add a detailed Appendix describing the experimental setup for the "Academic Paper Search" fine-tuning.

- Validate LLM-Generated Bandit Feedback (Strengthening) The framework relies on the premise that LLMs can provide useful "noisy bandit feedback signals". However, LLMs are known to struggle with self-evaluation and scalar reward estimation. If the feedback is too noisy, the search agent is effectively navigating blindly.

---

> ### Author Response · Authors · 2026-01-31
>
> We deeply appreciate the valuable feedback commented by the reviewer. We will answer your questions and concerns below. If you have any further questions or concerns, please let us know.
>
> > Concerns Regarding Synthetic Data Scale and Generalization: My primary concern lies in the complexity of the synthetic environments. The experiments primarily rely on small-scale settings, such as binary trees of depth 6 or 4x4 grids. While the paper claims generalization capabilities, the results in Appendix Figure 20 paint a different picture. There is a significant performance drop when the tree depth increases from 6 (seen) to 8 (unseen). This sharp degradation questions the core claim that the Transformer has truly "internalized" the recursive search algorithm. It suggests the model might be relying on shallow pattern matching or local heuristics that fail to scale, rather than learning a generalizable search procedure.
>
> > Clear Evidence of Generalization (with Limitations) The authors provide clear evidence that the learned behaviors are not merely memorized but generalized.
>
> > Contextualize Generalization Limits (Critical): While the paper claims the models can generalize, the results in Appendix Figure 20 show a clear performance degradation when the tree depth increases from 6 to 8. This suggests that the "internalized search" might be limited to a finite horizon or specific pattern matching rather than a fully recursive algorithm.
>
> > Request: Please explicitly discuss this degradation in the main text (Section 5 or Limitations). Analyze why the performance drops—is it due to context length limitations, attention dispersion, or a breakdown of the learned heuristic? The claims regarding generalization should be calibrated to reflect these empirical bounds (e.g., "generalizes within a local range" rather than broad generalization).
>
> We have toned down our claim regarding generalization, and added the limitations of our generalization analysis. Please see Section 5 of our revision. For your convenience, we highlight our revised sentences here.
>
> ```
> These empirical results show that our behavior-cloned Transformers present evidence of generalization to novel tasks. In particular, our Transformer models generalize well within a local range, whereas their generalization performance degrades over longer ranges. We conjecture that this behavior stems from practical limitations such as short context length and limited model capacity.
> ```
>
> > Lack of Detail and Motivation in Targeted Fine-Tuning (Chapter 6): The transition to the "Academic Paper Search" task in Chapter 6 feels somewhat disconnected from the rigorous synthetic analysis. It lacks crucial experimental details regarding the fine-tuning process. For instance, the size of the training dataset, the number of epochs are not clearly detailed in the main text. The motivation for this specific task setup needs strengthening. It is unclear why this specific domain was chosen as the primary validator for "real-world" performance over more standard reasoning benchmarks.
>
> Here, we would like to describe the motivation of targeted fine-tuning experiments. The following sentences have been added in Section 6 of the revision.
>
> ```
> We choose this problem because its state expansion and evaluation are more structured than in more generic reasoning tasks such as GSM8K (Cobbe et al., 2021) and HotpotQA (Yang et al., 2018). The Academic Paper Search problem provides a fixed set of state expansions, since each paper has a predetermined set of child papers, and supports semantic-based state evaluation, since the distance between two papers can be defined in a straightforward manner. Consequently, this setting aligns well with our formulation of unknown tree search with bandit feedback.
> ```
>
> For the details of these experiments, please see an answer below.
>
> > Provide Missing Details for Targeted Fine-Tuning (Critical) Section 6 demonstrates that fine-tuning Qwen3-8B improves search capabilities, but lacks critical reproducibility details.
>
> > Request: Please add a detailed Appendix describing the experimental setup for the "Academic Paper Search" fine-tuning.
>
> Thank you for pointing this out. We have updated Section E.2 accordingly. Please see Section E.2 of our revision.

---

> ### Author Response · Authors · 2026-01-31
>
> > Reliability of LLM-Generated Bandit Feedback: The real-world scenarios rely on the assumption that "we can prompt the LLM to directly estimate state values... These realizations serve as noisy bandit feedback signals". This introduces a critical unverified variable: How accurate are these LLM-generated values? If the "bandit feedback" is highly inaccurate or inconsistent, the search agent is effectively navigating with a broken compass. The paper focuses on the search policy but takes the reward signal for granted. A high search capability on top of unreliable value estimation does not yield a good solver. The paper should evaluate the correlation between the LLM's estimated values and the ground truth (or task success) to validate this setup.
>
> > Validate LLM-Generated Bandit Feedback (Strengthening) The framework relies on the premise that LLMs can provide useful "noisy bandit feedback signals". However, LLMs are known to struggle with self-evaluation and scalar reward estimation. If the feedback is too noisy, the search agent is effectively navigating blindly.
>
> We appreciate your insightful feedback. Consistent with prior work on LLM-as-a-Judge [1, 2], LLM-generated evaluations are not necessarily inaccurate or inconsistent, and many recent studies [3, 4, 5] have adopted LLM-as-a-Judge frameworks that produce scalar or bounded scalar reward estimates as an evaluation approach. However, our goal is to isolate and assess the search capabilities of LLMs or Transformers independent of any potential bias introduced by LLM-generated values. Therefore, validating the reliability of LLM-generated bandit feedback is beyond the scope of this work.
>
> [1] Li, Dawei, et al., From Generation to Judgment: Opportunities and Challenges of LLM-as-a-judge. EMNLP, 2025.
>
> [2] Zheng, Lianmin, et al., Judging LLM-as-a-Judge with MT-Bench and Chatbot Arena. NeurIPS, 2023. Datasets and Benchmarks Track.
>
> [3] Liu, Yang, et al., G-Eval: NLG Evaluation using GPT-4 with Better Human Alignment. EMNLP, 2023.
>
> [4] Kim, Seungone, et al., Prometheus: Inducing Fine-grained Evaluation Capability in Language Models. ICLR, 2024.
>
> [5] Zhang, Erica, et al., LLM-Lasso: A Robust Framework for Domain-Informed Feature Selection and Regularization. arXiv preprint arXiv:2502.10648, 2025.
>
> Thank you again for your productive feedback.

---

### Review · Reviewer_N5Wq · 2025-12-24

**Summary Of Contributions:**

The paper investigates whether Transformer architectures can internalize search algorithms to solve problems where the search space is not known a priori but revealed incrementally (unknown tree search with bandit feedback). The authors propose a simplified framework where tree extensions and feedback signals are externally specified, allowing for controlled evaluation of search capabilities. Theoretical analysis shows that Transformers are expressive enough to implement various search strategies and empirically they can learn them via behavior cloning. But off-the-shelf LLMs do not currently exploit this capacity well; targeted fine-tuning on search trajectories can substantially enhance their search behavior.

Pros:
- The paper introduces a novel "unknown tree search with bandit feedback" set up isolating the model's search selection ability
- Theorems 1 & 2 provide well-demonstration that Transformers with constant depth and specific embedding dimensions can exactly implement leaf-based and path-based search policies.
- Provide comprehensive empirical analysis and generalization experiments for training Transformers from scratch to clone various search algorithms, and generalize to unseen conditions.

Cons:
- The theoretical results are based on hard attention, the practical connection to actual multi-head softmax-based Transformers is unknown (which is acknowledged).
- The experiments are primarily on synthetic problems and academic paper search, which may not fully capture the complexity of real-world reasoning tasks. Generalization to longer sequences and larger problem instances remains challenging (which is acknowledged).

**Audience:**

Yes

**Audience Explanation:**

Understanding whether the Transformer architecture itself can internalize search loops is a core problem of LLMs’ ability. It will be interesting to broader community and research related to thinking, reasoning-focused, search-based inference, and LLM-as-planners, agentic behavior.

**Broader Impact Concerns:**

No major ethical concerns

**Claims And Evidence:**

Yes

**Claims Explanation:**

The claims are mostly well-supported
- Theoretical Claims: Theorems 1 & 2 provide explicit constructions and proofs to show that Transformers with constant depth and specific embedding dimensions can exactly implement leaf-based and path-based search policies.
- Empirical Claims: Comprehensive experiments are conducted, with clear metrics and comparisons to reference algorithms. Figures 7, 8, 19, and 20, show the testing on the models on unseen tree depths, step counts, and wall densities to support the generalization claim. But the claim of "generalizing to unseen conditions" is somewhat overstated given that generalization to deeper trees shows significant degradation (Figure 20). The improvement in LLM performance after targeted fine-tuning is also evaluated in Table 1, with comparisons to both pretrained LLMs and LLMs combined with external algorithms.

**Requested Changes:**

- Explicitly discuss the use of hardmax vs softmax, also whether multi-head attention or layer norm might help or hinder such constructions. Also it would be interesting to ablate the sensitivity of components of the Transformer architecture to the search abilities.
- Explicitly discuss tokenization assumptions.
- More explanation on the path-based KL divergence results.
- Provide at least one real world experiment could strengthen the paper.
- The fine-tuning experiments rely on trajectories generated by proprietary models like Gemini. This can potentially inject unknown biases and behaviors into the trained models. A short comment on this would be appropriate.

---

> ### Author Response · Authors · 2026-01-31
>
> We deeply appreciate the valuable feedback commented by the reviewer. We will answer your questions and concerns below. If you have any further questions or concerns, please let us know.
>
> > The theoretical results are based on hard attention, the practical connection to actual multi-head softmax-based Transformers is unknown (which is acknowledged).
>
> > Explicitly discuss the use of hardmax vs softmax, also whether multi-head attention or layer norm might help or hinder such constructions. Also it would be interesting to ablate the sensitivity of components of the Transformer architecture to the search abilities.
>
> We have added discussion on this topic in Section 4 of our revision. In addition, we have already provided the thorough discussion of our theoretical Transformer model in Section D.1. Please look into Sections 4 and D.1 of our revision. For the empirical analysis you mentioned, we think that training the Transformers with hardmax and without layer normalization is practically challenging. Thus, it is omitted in the current version.
>
> > The experiments are primarily on synthetic problems and academic paper search, which may not fully capture the complexity of real-world reasoning tasks. Generalization to longer sequences and larger problem instances remains challenging (which is acknowledged).
>
> > Provide at least one real world experiment could strengthen the paper.
>
> Our experiments including synthetic problems and academic paper search are designed to reflect our problem formulation of unknown tree search with bandit feedback. This formulation intentionally isolates the search capabilities of Transformers. For this reason, it is difficult to include real-world reasoning tasks since such tasks require defining states and evaluation functions using LLMs themselves. Instead of introducing additional real-world tasks, we expand the discussion of the motivation for targeted fine-tuning experiments. The following sentences have been added to Section 6 in the revised version.
>
> ```
> We choose this problem because its state expansion and evaluation are more structured than in more generic reasoning tasks such as GSM8K (Cobbe et al., 2021) and HotpotQA (Yang et al., 2018). The Academic Paper Search problem provides a fixed set of state expansions, since each paper has a predetermined set of child papers, and supports semantic-based state evaluation, since the distance between two papers can be defined in a straightforward manner. Consequently, this setting aligns well with our formulation of unknown tree search with bandit feedback.
> ```
>
> > But the claim of "generalizing to unseen conditions" is somewhat overstated given that generalization to deeper trees shows significant degradation (Figure 20).
>
> Thank you for pointing this out. We have toned down this claim. Please see Section 5 of our revision.
>
> > Explicitly discuss tokenization assumptions.
>
> We have revised Section 4 to include a detailed discussion of our tokenization schemes. Please refer to the revision.

---

> ### Author Response · Authors · 2026-01-31
>
> > More explanation on the path-based KL divergence results.
>
> Thank you for raising this issue. We added more thorough discussion of our KL divergence results in Section 5 of our revision. We would like to highlight the updated paragraph here; you can find this paragraph in Section 5.
>
> ```
> As shown in Figure 6, the leaf sampling algorithms achieve low KL divergence with their reference counterparts, suggesting successful behavior alignment. In contrast, for path sampling algorithms, they exhibit significantly higher KL divergence, compared to the leaf sampling algorithms. This suggests that they are unlikely to fully replicate the reference algorithms' behavior despite matching the performance on all of our metrics; the KL divergence results do not agree with the resulting performance reported in Figure 5. This discrepancy arises because of the following rationales: (i) the training traces for path-based methods do not explicitly encode the tree-policy path for each step; (ii) the randomness included in the process of particular path-based methods, such as the uniform path sampling, pure exploration policy-guided path sampling, and UCT policy-guided path sampling, makes KL divergence higher. Regarding the first rationale, the Transformer must infer the tree-based policy and select the next state in a single step, without intermediate tree traversal-related outputs. Due to this, it is a naturally challenging problem under our tokenization scheme. In addition, this aligns with our theoretical analysis in Section 4, where the trace format as described in Section C.1 explicitly requires the Transformer to output the tree traversal policy before selecting the next state. Regarding the second rationale, the order of some subsequent states may be interchangeable, since their ordering does not significantly affect the eventual outcome. This rationale is supported by two observations: (i) pure exploration policy-guided path sampling shows periodic KL divergence patterns with recurring low-divergence regions; (ii) greedy policy-guided path sampling yields lower KL divergence than other policy-guided path sampling methods.
> ```
>
> > The fine-tuning experiments rely on trajectories generated by proprietary models like Gemini. This can potentially inject unknown biases and behaviors into the trained models. A short comment on this would be appropriate.
>
> We have revised Section 6 accordingly. Please look into our revision.
>
> Thank you again for your productive feedback.

---

> > ### Comment · Reviewer_N5Wq · 2026-02-01
> >
> > Thank you for the detailed response and the corresponding revision, which have fully addressed my concerns.

---

### Review · Reviewer_ML1j · 2026-01-17

**Summary Of Contributions:**

The work analyzes whether transformers can implement tree-search strategies on two controlled benchmarks which implement unknown tree search with bandit feedback. The environment reveals the next plausible nodes given the current state and a noisy value estimate for each. The model then selects the next node from this list. The paper first shows explicit constructions showing constant-depth Transformers can exactly implement several selection policies. Then, the work shows behavior cloning results where small Transformers trained from scratch can match reference algorithms’s aggregate search metrics on two synthetic environments (tree-search and navigation) and generalize to some shifted conditions (more steps, deeper trees). The work also shows that fine-tuning improves performance. Overall, it studies selection policies under uncertainty, and it surfaces a methodological point that I really liked: matching metrics does not necessarily mean the model learned the underlying algorithm (kl divergence of next-state selection distributions indicate very different behavior).

**Audience:**

Yes

**Audience Explanation:**

Knowing whether LLMs can be enhanced using external search algorithms or whether they can internalize them is an interesting problem. The paper itself has several nice aspects:

The paper is methodologically strong in its decomposition of the search problem. By externalizing expansion and bandit feedback, the benchmark focuses directly on the selection problem and enables controlled comparisons across policies. It shows transformers have capacity to implement the search algorithms in these settings.

The experimental evaluation is also extensive, covering multiple metrics across both tree and maze environments.

I particularly appreciated the nuance in the KL divergence analysis. The work shows that path-based policies can match aggregate performance metrics while diverging significantly in their next-action distributions (high KL divergence). This is interesting evidence showing that models do not successfully internalize the specific algorithms but achieve similar outcomes.

**Claims And Evidence:**

Yes

**Claims Explanation:**

I worry that the paper’s framing overreaches the evidence provided.  However, if claims are adjusted it’s a nice analysis of the selection problem.

W1 The benchmark removes two of the most difficult components of search in LLM settings: generating candidates and evaluating the value of partial solutions. By providing these externally, the task is effectively reduced to supervised ranking or classification over an enumerated set. While the model performs selection -- claiming it has internalized the full search process seems to me like a strong claim.

W2 There is a tension between the paper’s core claim that Transformers do implement search algorithms and its own experimental results. The KL divergence analysis shows that for path-based strategies, the model produces action distributions that differ distinctively from the underlying algorithms. This undermines the claim that the model is implementing the target algorithm; it is simply achieving a similar outcome, likely through different heuristics.

W3 While the theoretical guarantees are technically correct, they feel weakly connected to the empirical results in this paper. The proofs rely on several idealizations, essentially saying that a Transformer can encode a lookup-table implementation of these algorithms. This does not offer much predictive power regarding how the models behave in the experiments, nor does it provide a way for the trained models to utilize the theoretical constructions to implement the underlying algorithm.

**Requested Changes:**

Address the above weaknesses.

---

> ### Author Response · Authors · 2026-01-31
>
> We deeply appreciate the valuable feedback commented by the reviewer. We will answer your questions and concerns below. If you have any further questions or concerns, please let us know.
>
> > I worry that the paper’s framing overreaches the evidence provided. However, if claims are adjusted it’s a nice analysis of the selection problem.
>
> Thank you for raising this issue. We have adjusted our claims by following your comments. Please look into our revision.
>
> > W1 The benchmark removes two of the most difficult components of search in LLM settings: generating candidates and evaluating the value of partial solutions. By providing these externally, the task is effectively reduced to supervised ranking or classification over an enumerated set. While the model performs selection -- claiming it has internalized the full search process seems to me like a strong claim.
>
> We have toned down our claim stating that our Transformers approximate the internalization of the search algorithm. Please see the abstract and Sections 1, 5, and 7 of our revision.
>
> > W2 There is a tension between the paper’s core claim that Transformers do implement search algorithms and its own experimental results. The KL divergence analysis shows that for path-based strategies, the model produces action distributions that differ distinctively from the underlying algorithms. This undermines the claim that the model is implementing the target algorithm; it is simply achieving a similar outcome, likely through different heuristics.
>
> > I particularly appreciated the nuance in the KL divergence analysis. The work shows that path-based policies can match aggregate performance metrics while diverging significantly in their next-action distributions (high KL divergence). This is interesting evidence showing that models do not successfully internalize the specific algorithms but achieve similar outcomes.
>
> Thank you for raising this issue. We added more thorough discussion on our KL divergence results in Section 5 of our revision. We would like to highlight the updated paragraph here; you can find this paragraph in Section 5.
>
> ```
> As shown in Figure 6, the leaf sampling algorithms achieve low KL divergence with their reference counterparts, suggesting successful behavior alignment. In contrast, for path sampling algorithms, they exhibit significantly higher KL divergence, compared to the leaf sampling algorithms. This suggests that they are unlikely to fully replicate the reference algorithms' behavior despite matching the performance on all of our metrics; the KL divergence results do not agree with the resulting performance reported in Figure 5. This discrepancy arises because of the following rationales: (i) the training traces for path-based methods do not explicitly encode the tree-policy path for each step; (ii) the randomness included in the process of particular path-based methods, such as the uniform path sampling, pure exploration policy-guided path sampling, and UCT policy-guided path sampling, makes KL divergence higher. Regarding the first rationale, the Transformer must infer the tree-based policy and select the next state in a single step, without intermediate tree traversal-related outputs. Due to this, it is a naturally challenging problem under our tokenization scheme. In addition, this aligns with our theoretical analysis in Section 4, where the trace format as described in Section C.1 explicitly requires the Transformer to output the tree traversal policy before selecting the next state. Regarding the second rationale, the order of some subsequent states may be interchangeable, since their ordering does not significantly affect the eventual outcome. This rationale is supported by two observations: (i) pure exploration policy-guided path sampling shows periodic KL divergence patterns with recurring low-divergence regions; (ii) greedy policy-guided path sampling yields lower KL divergence than other policy-guided path sampling methods.
> ```
>
> > W3 While the theoretical guarantees are technically correct, they feel weakly connected to the empirical results in this paper. The proofs rely on several idealizations, essentially saying that a Transformer can encode a lookup-table implementation of these algorithms. This does not offer much predictive power regarding how the models behave in the experiments, nor does it provide a way for the trained models to utilize the theoretical constructions to implement the underlying algorithm.
>
> We have updated our manuscript by following your comment. Also we highlighted the discussion of this theoretical analysis. Please see Section 4 of our revision.
>
> Thank you again for your productive feedback.

---

> > ### Comment · Reviewer_ML1j · 2026-03-02
> >
> > Thanks for the rebuttal, my major concerns were satisfactorily addressed!

---

### Author Response · Authors · 2026-01-31

We thank the reviewers' constructive comments. We would like to highlight that all reviewers agreed that the claims made in the submission are supported by accurate, convincing, and clear evidences, and at least some individuals in TMLR's audience would be interested in knowing the findings of this paper.

In the revised version, red text indicates edits, and blue text indicates highlighting. Here we provide the summary of our revision.

- Revised the abstract to tone down our claims about the internalization and generalization of search algorithms.
- Revised Section 1; similarly we have toned down our claims.
- Added a description of our tokenization schemes in Section 3.3.
- Mentioned our model assumptions in Section 4.
- Moved the details of our tokenization schemes to Section 4.
- Made minor revisions to Theorem 2.
- Mentioned that the Transformers' parameters proved in Theorems 1 and 2 are ideal parameters.
- Discussed the results of KL divergence more thoroughly in Section 5.
- Made minor revisions to Finding 2.
- Described the motivation of the Academic Paper Search problem in Section 6.
- Made minor revisions to Finding 3.
- Revised Section 7.
- Added the details of fine-tuning for enhancing search capabilities of LLMs in Section E.2.
- Made more minor revisions to various locations.

Thank you again for your efforts to provide the valuable reviews.

---

### Decision · Action_Editor_hBSZ · 2026-03-02

**Recommendation:** Accept as is

**Audience:**

Yes

**Audience Explanation:**

The paper’s combination of a formal setup, exact representational results for Transformers, and empirical tests of learned search behavior makes it relevant to multiple TMLR sub-communities.

**Claims And Evidence:**

Yes

**Claims Explanation:**

Based on the reviewers’ recommendations and the authors’ revisions, I recommend acceptance.

The paper is technically strong and well executed. A key strength is the clearly defined methodological setup (“unknown tree search with bandit feedback”), which isolates the model’s search-selection ability in a way that supports meaningful analysis. The theoretical component is also convincing: the stated results (including Theorems 1 and 2) provide a rigorous demonstration of how Transformers with constant depth and appropriate embedding dimensions can exactly implement both leaf-based and path-based search policies. This gives the work a solid formal foundation rather than relying only on empirical evidence.

The empirical section is another major strength. The paper includes extensive experiments on training Transformers from scratch to imitate different search algorithms, along with evaluations under unseen conditions. The rebuttal/revision appears to have substantially improved the manuscript by clarifying experimental details (including academic search fine-tuning parameters), tempering claims about generalization, and better motivating the structured state expansion design choices. These changes address the primary concerns raised in review and improve the paper’s transparency and positioning.

I also agree with the reviewers that the remaining limitations are real and should be acknowledged: in particular, the observed weakness in generalization to unseen depths suggests that the learned behavior may still depend partly on local patterns rather than a fully depth-invariant internal procedure. However, this limitation is appropriately discussed and does not undermine the paper’s core technical contributions or the validity of its conclusions within scope.

Overall, the paper presents a rigorous, methodologically careful, and technically sound contribution, with strong theory and comprehensive experiments. With the revised claims and clarifications, it meets the bar for publication in TMLR.